# Meta-QTL Analysis and Genes Responsible for Plant and Ear Height in Maize (*Zea mays* L.)

**DOI:** 10.3390/plants14131943

**Published:** 2025-06-24

**Authors:** Xin Li, Xiaoqiang Zhao, Siqi Sun, Kejin Tao, Yining Niu

**Affiliations:** 1State Key Laboratory of Aridland Crop Science, Gansu Agricultural University, Lanzhou 730070, China; m18214360633@163.com (X.L.); sunsiqi1215@163.com (S.S.); 2State Key Laboratory of Herbage Improvement and Grassland Agro-Ecosystems, College of Pastoral Agriculture Science and Technology, Lanzhou University, Lanzhou 730000, China; taokj2024@lzu.edu.cn

**Keywords:** maize, plant height, ear height, Meta-QTL, meta-analysis, candidate genes

## Abstract

Plant height (PH) and ear height (EH) are closely related to dense planting characteristics and lodging resistance of maize (*Zea mays* L.). Increasing the planting density will lead to changes in the structural characteristics of maize plants, such as reduced stem length and stem strength, thereby influencing their yield and quality. Therefore, analyzing the genetic basis of PH and EH in maize can provide valuable information for cultivating ideal plant types with suitable PH and EH. This study aims to identify stable genomic regions and candidate genes associated with PH and EH in maize through Meta-QTL (MQTL) analysis. A total of 187 original QTLs were collected from 13 published articles on QTL localization related to maize PH and EH. A high-density consistency map with a total length of 6970.00 cM was constructed, and 152 original QTLs were successfully projected into the consistency map. The remaining 35 QTLs could not be projected onto the consistency map, which may be attributed to a lack of common markers between the original and consistency map or to the QTL exhibiting low phenotypic variance explained (PVE), resulting in large confidence intervals (CIs). Then, 29 MQTLs were identified on 10 chromosomes via meta-analysis. Among them, the three identified MQTLs, i.e., MQTL4-1, MQTL4-2, and MQTL6-1, were specifically controlled by maize EH. Further analysis achieved 188 candidate genes in all MQTL intervals, which were related to maize plant development and morphogenesis. Meanwhile, the gene ontology (GO) enrichment analysis revealed that these candidate genes were involved in 77 GO annotations. These findings thus will help us better understand the molecular genetic basis of maize PH and EH under various environments, and thereby achieve an increased yield with maize dense planting breeding.

## 1. Introduction

Maize (*Zea mays* L.) is one of the major food crops in the world; it is widely used in food, feed, and industrial raw materials [1,2]. In recent years, increasing the yield of maize has become an important breeding goal, as we are confronted with the challenges, including rapid population growth, the decrease of cultivated land area, and various environmental stresses. Research has found that the yield of maize is affected by multiple factors such as the length of daylight, soil type, planting density, temperature, and nutrient supply. Among them, optimizing planting density serves as an effective strategy and a key approach to achieving consistently high yields in maize cultivation [3]. In China, maize yield has consistently increased alongside rising planting densities, growing from less than 30,000 plants/ha in 1949 to the current range of 49,850 to 65,180 plants/ha, and the yield per unit area has shown a remarkable 6.6-fold increase [4,5]. But this is far lower than the planting density of 82,230–92,100 plants/ha in the USA, which is the world’s largest producer of maize [6], thereby indicating that the maize planting density in China could be increased greatly.

However, increasing the planting density will significantly change the structural characteristics of maize plants, leading to the elongation and thinning of the stem and a decrease in mechanical strength, subsequently affecting the plant’s ability to resist lodging and ultimately resulting in a reduction in yield [7,8]. Therefore, studying the genetic basis of maize plant type traits can provide valuable information for novel varieties with density resistance and high yield.

Maize plant type traits, including plant height (PH), ear height (EH), leaf length, leaf width, and leaf Angle [9,10]. Among them, PH and EH are the characteristic traits that affect the structure of maize plants [11]. On the one hand, appropriately reducing the PH and the EH of maize can increase its planting density and yield. Meanwhile, they can also improve the lodging resistance of maize and facilitate mechanical harvesting [12]. Moreover, excessive PH and EH are important limiting factors for dense maize planting. While PH and EH of maize are complex quantitative traits that are consistently regulated by multiple genes and environmental factors. In recent years, with the rapid development of molecular marker technology, the source of genetic variation in PH and EH has been explored using quantitative trait locus (QTL) detection in bi-parental maize, mapping populations in multiple studies [13,14]. For example, six PH-related QTLs and 15 EH-related QTLs in the F_2:3_ families and recombinant inbred line (RIL) population were identified via single nucleotide polymorphism (SNP) genetic map and composite interval mapping (CIM) method and further identifying three major genomic regions for PH and EH in bin1.05, bin5.04–5.05, and bin6.04–6.05 (Li et al. [15]). Using double haploid (DH) population, two PH-related QTLs (explained phenotypic variations [PVE], 9.70% and 13.78%, respectively) and one EH-related QTL (PVE 18.09%) were mapped by simple sequence repeats (SSR) markers (Choi et al. [16]). 30 QTLs responsible for PH and EH were detected from across RIL populations with, which were constructed by Yu82 × Yu87-1, Yu82 × Shen137, Zong3 × Yu87-1 and Yu537 × Shen137 hybrids, respectively, and these detected QTLs were distributed on chromosomes 1, 2, 3, 4, 5, 6, 7, and 10 (Ku et al. [17]). However, due to differences in the size of mapping populations, genetic background, and selected markers, the same QTL is rarely found in different studies, which severely restricts the application potential of QTLs in maize molecular marker-assisted selection (MAS) breeding.

Fortunately, Meta-QTL (MQTL) analysis can conduct a comprehensive analysis of original QTLs from different studies and then screen out QTLs with lower confidence intervals (CI), high consistency, and greater influence on the target traits [18]. Among them, a lower CI can improve the positioning accuracy, reduce the number of candidate genes, and enhance the reliability of the results. At present, this method has been widely used in genetic investigation and candidate gene identification of complex quantitative traits in plant species. In maize, Sheoran et al. [19] integrated original 542 QTLs from 33 published studies on tolerance to different abiotic stresses (drought, heat, salinity, cold, and waterlogging), who further performed MQTL analysis to obtain 32 MQTLs associated with multiple tolerance, and these MQTL regions identified 1907 potential candidate genes. Tang et al. [20] conducted MQTL analysis of 697 original QTLs related to maize quality traits and finally found 40 MQTLs. Li et al. [21] collected 511 original QTLs related to maize grain yield composition and flowering time, resulting in 83 MQTLs being identified. In Soybean, Yin et al. [22] conducted MQTL analysis using 182 original QTLs related to soybean PH and obtained 36 MQTLs. Within these MQTL intervals, 4259 candidate genes were identified, among which 40 were involved in plant growth and stem elongation. It can be seen from this that the use of MQTL analysis can help us determine the candidate genes related to maize PH and EH development.

Revealing the genetic basis of PH and EH can lay a foundation for cultivating density-tolerant varieties and increasing the yields of maize. Based on the above considerations, the aims of the present study were to integrate multiple previously published QTLs related to PH and EH in maize, identify crucial genomic regions controlling these traits through MQTL analysis, obtain the corresponding MQTLs, and predict candidate genes. These findings will help us further understand the genetic basis of maize PH and EH, thereby cultivating varieties suitable for dense planting and increasing yield in maize.

## 2. Results

### 2.1. Integration of QTL-Related Information of PH and EH in Maize

Through comprehensive bioinformatics analysis, we systematically collected and analyzed 13 original research articles for nearly 15 years from the databases, namely NCBI (https://www.ncbi.nlm.nih.gov/; accessed on 23 October 2024), CNKI (https://www.cnki.net/; accessed on 23 October 2024), and maizeGDB (http://www.maizegdb.org/; accessed on 23 October 2024). For the 13 original research, the population sizes varied from 162 to 271 individuals, including six F_2_, six F_2:3_, one F_4,_ three RIL, and one DH populations; and their genetic map length were constructed by SSR, amplified fragment length polymorphisms (AFLP), and insertion-deletion (Indel) markers, with the total length varying from 985.71 to 4734.51 cM. Moreover, the phenotypes of PH and EH from these populations were evaluated in 70 environments. Subsequently, the QTL analysis identified 83 original QTLs responsible for PH and 104 original QTLs related to EH (Table 1).

For these original QTLs, which are distributed on all ten maize chromosomes, accounting for 17.65%, 7.49%, 19.79%, 10.16%, 9.09%, 3.21%, 8.56%, 8.56%, 3.74%, and 11.76%, respectively. The maximum number of QTLs for PH and EH were located on chromosome 3 (approximately 10.16% and 9.63%, respectively), while chromosome 6 had the least number of PH and EH QTLs (approximately 1.07% and 2.14%, respectively) (Figure 1A). Meanwhile the PVE of single QTL for PH and EH was 4.11–31.00% and 3.23–31.93%, average values were 10.41 and 10.76, respectively, and the LOD values varied from 2.50 to 18.50 for PH and 2.50 to 20.47 for EH, average values were 5.11 and 5.35, respectively (Figure 1B,C). Moreover, there were only six PH-QTLs and four EH-QTLs that displayed high logarithm of odds (LOD) threshold (LOD > 10), respectively (Figure 1B), and only 24.59% of PH-QTLs and 32.62% EH-QTLs showed high PVE, respectively (Figure 1C). It is thus necessary to identify novel major QTLs with high values of LOD and PVE in the future.

### 2.2. Construction and QTL Projection of Consensus Map of Maize PH and EH

We then used the BioMercator v4.2.3 software (https://urgi.versailles.inra.fr/Tools/BioMercator-V4; accessed on 20 December 2024) and referred to the IBM2 2008 neighbors reference map to successfully develop a comprehensive consensus map, i.e., the total length of consensus map was 6970.00 cM, containing a total of 901 markers, with an average interval of 7.68 cM between markers. In addition, the 152 original QTLs (approximately 81.28%) were well mapped into the consensus map (Figure 2).

### 2.3. Meta-QTL Analysis and Candidate Gene Identification in MQTL Region

Through meta-analysis, a total of 29 MQTLs were identified on 10 chromosomes, six on chromosome 1, three on chromosome 2, six on chromosome 3, two on chromosome 4, one on chromosome 5, one on chromosome 6, two on chromosome 7, three on chromosome 8, one on chromosome 9, and four on chromosome 10. The physical distance of a single MQTL varied from 0.04 to 30.08 Mb. Meta-analysis was able to significantly reduce the CI of the original QTL, with the CI range of the 29 identified MQTLs ranging from 0.80 to 75.20 cM and the mean distance being 26.27 cM. Each MQTL contained 2–10 original QTLs. Among them, the CI of MQTL2-1 was very narrow, less than 1 cM, and its CI was 0.8 cM. At the same time, we also detected that 20 MQTLs were commonly controlling PH and EH, i.e., MQTL1-1 (umc2546-umc1292; Bin1.01), MQTL1-2 (bnlg1178-bnlg1429; Bin1.02), MQTL1-3 (umc1073-bnlg1203; Bin1.03), MQTL1-6 (bnlg1643-umc1991; Bin1.08), MQTL2-1 (umc1542-umc1227; Bin2.01–2.02), MQTL2-2 (umc1961-mmc0111; Bin2.02), MQTL2-3 (bnlg1329-umc2129; Bin2.07), MQTL3-1 (bnlg1647-bnlg1904; Bin3.02–3.04), MQTL3-2 (umc1655-umc1504; Bin3.04), MQTL3-3 (umc18397-umc1087; Bin3.04–3.05), MQTL3-4 (umc1644-umc2269; Bin3.06), MQTL3-5 (umc1489-umc1286; Bin3.07), MQTL3-6 (umc1502-umc2084; Bin3.09–3.10), MQTL5-1 (umc2305-bnlg1346; Bin5.06–5.07), MQTL7-1 (umc1666-umc1932; Bin7.02), MQTL7-2 (umc2329-bnlg1666; Bin7.03–7.04), MQTL8-2 (bnlg666-umc2210; Bin8.05), MQTL9-1 (bnlg1626-bnlg1209; Bin9.03–9.04), MQTL10-1 (umc1666-umc1932; Bin10.01–10.02), and MQTL10-3 (bnlg1079-umc1938; Bin10.03–10.04), suggesting that the Bin1.01, 1.02, 1.03, 1.08, 2.01–2.02, 2.02, 2.07, 3.02–3.04, 3.04, 3.04–3.05, 3.06, 3.07, 3.09–3.10, 5.06–5.07, 7.02, 7.03–7.04, 8.05, 9.03–9.04, 10.01–10.02 and 10.03–10.04 were important regions commonly regulating the PH and EH of maize plant, these MQTLs intervals may exist crucial candidate gene for determining maize plant type plasticity (Table 2).

Moreover, the detected 29 MQTLs were further projected on the physical map B73 RefGen_V4 (https://www.maizegdb.org/genome/assembly/B73%20RefGen_v3; accessed on 26 December 2024). to identify potential candidate genes regulating maize plant type structure. The results showed that 188 candidate genes were obtained in 27 MQTL regions (no candidate genes in MQTL2-1 and MQTL2-2) (Figure 3), which might be the result of a false-positive original QTL mapping from previous experiments. Additionally, these identified candidate genes were unevenly distributed on the ten chromosomes of maize. Among them, the number of candidate genes on chromosome 1 was the largest, with 49, followed by chromosome 5. Next, to reveal the functional classification of these candidate genes, we conducted gene ontology (GO) enrichment analysis on 188 candidate genes identified within 27 MQTL intervals. The GO enrichment analysis indicated that the 188 candidate genes were mainly classified into biological processes (47 items), cellular components (19 items), and molecular functions (11 items) (Figure 4). In biological processes, it is mainly enriched in “cell wall organization or biogenesis”, “cell wall organization”, and “external encapsulating structure organization”. The most abundant GO terms related to cellular components were “extracellular region”, “photosystem I”, “photosystem II”, “cytoskeleton”, and “actin cytoskeleton”. “Xyloglucosyl transferase activity”, “chlorophyll binding”, and “glucosyltransferase activity” were enriched in the molecular functions annotation. Thereby these targeted candidate genes were involved in multiple metabolic processes, including biosynthesis and signal transduction of plant hormones (abscisic acid [ABA], auxin [IAA], cytokinin [CTK], ethylene [ETH], gibberellin [GA], and brassinosteroid [BR]; approximately for 32.09%), metabolism of phenylpropanoid, cellulose synthesis, sugars accumulation, cell division and growth (approximately for 32.62%), and photosynthesis (approximately for 9.09%), (Appendix A).

## 3. Discussion

With rapid population growth, people’s demand for food is also constantly increasing, so increasing food production has become a prominent challenge facing the world. At present, maize in China accounts for 17.4% of the global total planting area, while its grain yield is only 6.3 t/ha, which is clearly lower than 11.1 t/ha in the United States [35]. The reason for this phenomenon is the significant increase in plant density per unit area and improvement of per-unit yield of maize. But we must also know that a higher planting density may have negative impacts on the growth of individual maize plants, including restricting leaf growth and expansion, decreasing photosynthetic performance [5], insufficient grain filling [36], thinning of plant stems and appearing severe lodging [4], subsequently resulting in large maize yield loss. PH and EH are the important agronomic traits closely related to the cultivation tolerance of high-density maize. Therefore, an in-depth investigation of the genetic basis of PH and EH will be conducive to developing new maize varieties with a dwarf plant and high density tolerance.

PH and EH are quantitative traits, controlled by major genes and multiple micro-effect genes, and are susceptible to environmental influences. Previous studies had shown that the main genetic modes of PH and EH were additive and partially dominant effects, with additive effects being dominant [37,38,39]. In addition, cytoplasmic effects and nuclear superiority also have potentially significant contributions to phenotypic variations [25]. Up to now, with the quick development of molecular marker technology, in particular, using QTL mapping and genome-wide association studies (GWAS) analysis, hundreds of genetic loci related to PH and EH have been identified [11,40,41]. Likewise, limited key genes regulating PH and EH have been localized and cloned via QTL fine mapping and mutants creation, including *vanishing tassel2* (*VT2*) [42], *sparse inflorescence1* (*SPI1*) [43], *brachytic2* (*BR2*)*,* and *brevis plant1* (*BV1*) [44,45], that were major involved in auxin biosynthesis, transport or signal transduction. However, due to multiple differences in population types, population sizes, marker types, and QTL methods utilized in previous studies, it is difficult to identify consistently or stably major QTLs for PH and EH, which severely limits their breeding application potential.

MQTL analysis is a method that can integrate the initial QTL data from different studies for a more comprehensive analysis. Meta-analysis can break through the limitations of various factors and solve the difficulties that make it difficult to locate all the target QTLs in a single QTL localization, and the estimated PVE of QTL is often unreliable. In the present study, we conducted an MQTL analysis on 187 original QTLs related to PH and EH from 13 different studies. A total of 29 MQTLs were identified on 10 chromosomes, and these MQTLs contained a total of 132 original QTLs. Among them, there were the largest number of MQTLs on chromosomes 1 and 3, suggesting that the two chromosomes are important chromosomal regions that regulate maize dense planting traits of PH and EH. Furthermore, it is noted that MQTLs identified on chromosomes 4 and 6 were only related to EH. We thus pay attention to these chromosomal regions to find potential target genes for improving maize EH in the future.

We further projected all detected MQTLs into the B73 RefGen_V4 physical map to achieve potential candidate genes regulating maize plant type development. Ultimately, we identified 188 target genes in these MQTL intervals (Appendix A), of which, 60 detected candidate genes within 27 MQTL intervals were related to the biosynthesis and signal transduction of six plant hormones of IAA, ABA, CTK, ETH, GA, and BR, suggesting that plant hormones, as a kind of signaling molecule, play an important role in the development process of maize plants. However, as an important signaling substance regulating the growth and development of maize, the synthesis and transduction mechanisms of IAA have not been fully clarified [46]. In this study, we identified 18 candidate genes related to IAA synthesis and signal transduction associated with the development of PH and EH in maize. This discovery can provide certain references for the subsequent clarification of the synthesis and transduction mechanisms of IAA. Additionally, *Zm00001d010247* encoding phenylalanine ammonia-lyase (PAL), was identified in MQTL8-1, which was related to the biosynthesis of lignin [47]. Appropriate lignin can maintain the stability of the cell wall, but if the content is too high, it may limit the elongation of cells, thereby affecting the growth of maize. In addition, studies have shown that increasing the cellulose content can enhance the thickness and flexibility of the stem wall [48], thereby enhancing the strength and anti-lodging ability of the stem [49,50], to enable the normal development of PH and EH. Fortunately, we identified four genes related to cellulose synthesis, i.e., *Zm00001d032776* (cellulose synthase 10), *Zm00001d032909* (cellulose synthase-like protein D3), *Zm00001d005775* (cellulose synthase A catalytic subunit 7), and *Zm00001d046691* (cellulose synthase A catalytic subunit 5). Carbohydrates in plants are the main carbon source; key structural elements responsible for PH and EH, but also participate in various physiological metabolic processes such as osmotic regulation and material transport. Among MQTL1-2, MQTL1-3, MQTL1-6, and MQTL7-1, we identified four candidate genes related to sugar biosynthesis and transport. They are, respectively, sucrose transporter 1 (*Zm00001d027854*), sucrose synthase 3 (*Zm00001d028508*), glucose transporter 3 (*Zm00001d032906*), and β-amylase (*Zm00001d019756*).

Plant development is inseparable from the division and elongation of cells. Among the 27 MQTL intervals, 21 mapped candidate genes were distributed in five MQTL intervals of MQTL1-6 (seven), MQTL3-2 (one), MQTL5-1 (11), MQTL9-1 (one), and MQTL10-4 (one); they belonged to expansin genes, i.e., ten expansin A (EXPA) and 11 expansin B (EXPB). Research has found that when *GmEXPB2* is overexpressed in soybeans, it can promote the overall growth of the plants [51]. In addition, photosynthesis provides a carbon source as a key building material [52]. Interestingly, approximately 9.09% of the candidate genes were related to photosynthesis. For example, *Zm00001d028434* encodes cryptochromine-1 (CRY1), which is a blue light receptor. In maize, when seedlings were under two light conditions (13 µmol m^−2^ s^−1^ and 70 µmol m^−2^ s^−1^), the overexpression of the *ZmCRY1b* gene significantly inhibited their yellowing and shading responses, thereby leading to a decrease in PH and EH [53]. *Zm00001d028472* is a far-red impaired responsive (*FAR1*) family protein. Tang et al. [54] found that the expression level of *ZmFAR1-3* was relatively high in the stem tissue of maize B73, suggesting that *FAR1* may regulate maize PH development. Actin depolymerization factor (ADF) family proteins are a type of actin-binding protein that can promote the polymerization of actin. In this study, we also identified five *ADF* genes, e.g., *Zm00001d028392*, *Zm00001d029656*, *Zm00001d051388*, *Zm00001d017516,* and *Zm00001d021497*. The *ADF* controlled cell elongation by regulating F-actin tissue, thereby promoting plant growth [55]. Meanwhile, *Zm00001d042315, Zm00001d023736,* and *Zm00001d024389* were *MADS-box* family, which controlled various developmental processes of flowering plants [56].

In summary, we have tried to construct a molecular network that may regulate the PH and EH development in maize (Figure 5). Briefly, when maize plants are growing rapidly, multiple metabolic pathways controlled by related genes will be triggered. For example, when triggering metabolic pathways related to the synthesis and signal transduction of plant hormones, it will promote or inhibit the elongation of stem cells. When triggering pathways related to phenylpropanoid metabolism and cellulose formation, they will affect the formation of the cell wall. Ultimately, under the combined influence of these pathways, the stem development is affected, resulting in significant differences in PH and EH from various maize materials.

## 4. Materials and Methods

### 4.1. Literature Search and Information Collection for PH and EH QTL Localization of Maize

The keywords of “maize”, “PH”, “EH”, and “QTL” were inputted three public databases, i.e., NCBI (https://www.ncbi.nlm.nih.gov/; accessed on 23 October 2024), CNKI ((https://www.cnki.net/; accessed on 23 October 2024), and maizeGDB (http://www.maizegdb.org/; accessed on 23 October 2024), respectively, subsequently collecting corresponding QTL information of PH and EH, including original chromosome location, CI, PVE, LOD value, parents name, population type, population size, and marker type. The collected QTL were subjected to a quality check, and QTL with PVE < 1% or LOD < 1.5 were removed. Among them, the CI and PVE represented two fundamental parameters for QTL characterization. The meta-analysis of QTL was mainly achieved through the QTL LOD score, PVE, position, and CI. When the original QTL CI was missing, we can calculate the 95% CI based on the population type, size, and PVE, with the equation provided by Darvasi and Soller (1997) [57], as follows:(1)$$CI=530/\left(N\times PVE\right)$$(2)$$CI=163/\left(N\times PVE\right)$$
where CI was the confidence interval of a QTL, N was the size of the mapping population, and PVE was the phenotypic variance explained by QTL. Equation (1) was applicable to backcross and F_2_ populations. Equation (2) was applied to RIL populations.

### 4.2. Consensus Map Construction and QTL Projection

The IBM2 2008 Neighbors map (http://www.maizegdb.org/; accessed on 22 November 2024) is a high-density genetic linkage map spanning 8054.28 cM. This comprehensive map incorporates multiple marker types, including restriction fragment length polymorphism (RFLP), SSR, random amplified polymorphic DNA (RAPD), and gene-based markers, and can integrate QTL information from different sources [18]. In this study, we compared the original maps of QTLs with the IBM2 2008 Neighbors reference map. Next, we import all the information of these QTLs, especially the starting and ending positions of CI, into the program and map them to their own linkage maps. Then, based on their relative distances from the flanking framework markers, they are further projected onto the reference map while maintaining a constant distance between the frame marks. Then, the homogeneous functions provided by the BioMercator (V4.2) software (https://urgi.versailles.inra.fr/Tools/BioMercator-V4; accessed on 22 December 2024) were used to transfer each QTL to the reference map to create a consensus map of QTL related to maize PH and EH. If the order of the side labels of the given QTL is reversed from that of the reference map, it has no effect on the projection of the QTL. If multiple QTLs are linked to the same marker, the CI is calculated using the formula proposed by Darvasi and Soller [57]. Otherwise, QTL is not predicted.

### 4.3. Meta-QTL Analysis of Maize PH and EH

We adopted the QTL Maps-analysis module in BioMercatorv4.2 software (https://urgi.versailles.inra.fr/Tools/BioMercator-V4; accessed on 22 December 2024) for MQTL analysis to determine the precise location of MQTL and its corresponding CI. Specifically, in accordance with the software specification and using the integrated CI data of QTL, we conducted a separate meta-analysis of PH-/EH-QTL using the chromosome walk method. Furthermore, when conducting meta-analysis, first based on the optimal model values, i.e., AIC (Akaike information content), AICc (AIC correction), AIC3 (AIC 3 candidate models), BIC (Bayesian information criterion), and AWE (average weight of evidence) were used to determine the number of potential MQTL on each chromosome in different experiments. The QTL model with the lowest value among at least three of the five models was used to determine the number of MQTLs on each chromosome [58,59]. Finally, the location and CI of the existence of MQTL were estimated by using the maximum likelihood ratio of Gauss’s theorem.

### 4.4. Identification and Functional Annotation of Candidate Genes in the MQTLs Interval

To identify candidate genes, the MaizeGDB website (https://www.maizegdb.org/; accessed on 25 December 2024) was used to determine the physical location of the flanker markers. If the flanker’s physical location cannot be found, the next nearest external marker is used to detect the genomic coordinates of the MQTL. Next, based on the physical length of the obtained MQTL, candidate genes related to plant development are retrieved from the maizeGDB database. Specifically, we used the “qTeller” tool provided on maizeGDB to select the expression dataset of B73 RefGen_v4 for plant development only, in order to identify the genes existing within the MQTL physical intervals [60]. Finally, for the identified candidate genes, we used the AmiGO 2 database (http://amigo.geneontology.org/amigo/; accessed on 5 March 2025) for GO enrichment analysis to understand the biological functions of MQTL.

## 5. Conclusions

In this study, an MQTL analysis was conducted using 187 integrated original QTLs related to maize PH and EH, and a total of 29 stable MQTLs were successfully identified; meanwhile, each MQTL interval contained 2–10 original QTLs. Among them, the largest number of MQTLs were detected on chromosomes 1 and 3, indicating that these two chromosomes may be important genomic regions controlling plant growth and development. Then, based on the identified MQTL physical distances, we identified 188 potential candidate genes in these MQTL intervals, which were involved in multiple metabolic processes, including six plant hormones biosynthesis and signal transduction, phenylpropanoid metabolism, cellulose formation, sugars accumulation, cell division and expansion, and photosynthetic capacity. Furthermore, based on the above results and previous studies, we constructed a possible molecular network related to the development of PH and EH in maize (Figure 5). The MQTLs and their corresponding candidate genes identified in this study can thus be used for MAS in future breeding strategies.

## Figures and Tables

**Figure 1 plants-14-01943-f001:**
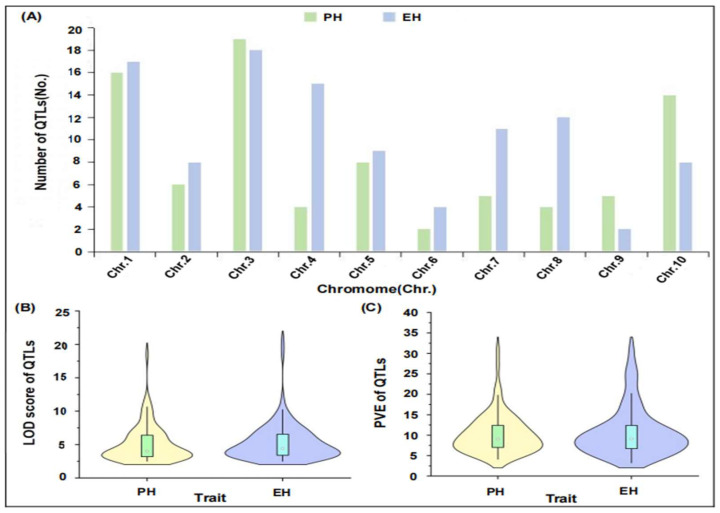
Integrated analysis of QTL information for maize PH and EH. (**A**) Chromosomal distribution of QTLs associated with PH and EH across 10 maize chromosomes; (**B**) LOD score value of identified QTLs for both traits; (**C**) Percentage of PVE by individual QTLs for PH and EH.

**Figure 2 plants-14-01943-f002:**
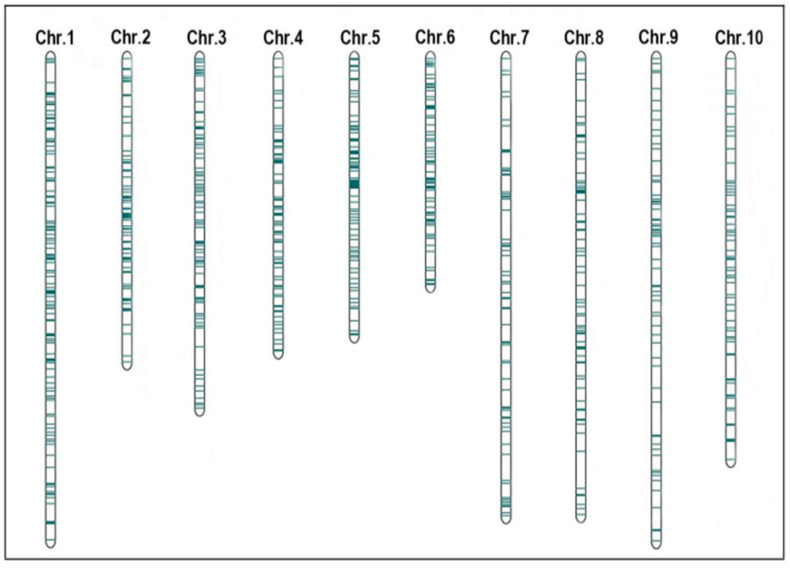
Consensus maps built on thirteen original map datasets.

**Figure 3 plants-14-01943-f003:**
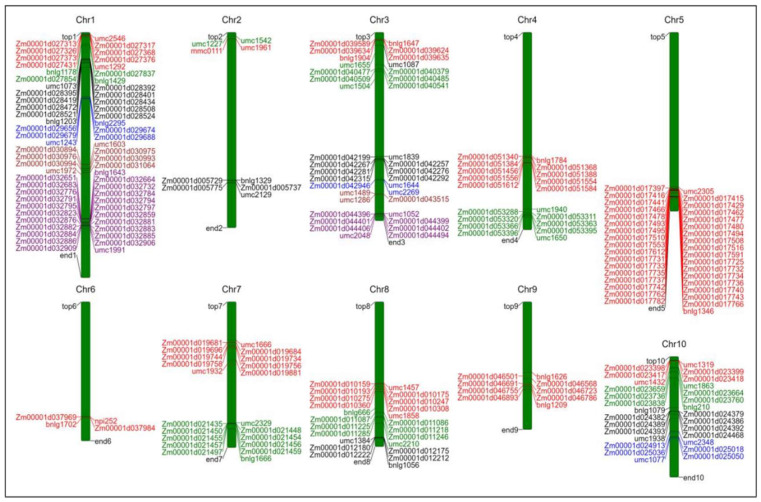
The localization map of candidate genes on 10 chromosomes. On each chromosome, different colors represent the candidate genes identified in different MQTL intervals and the marker intervals where the MQTL is located.

**Figure 4 plants-14-01943-f004:**
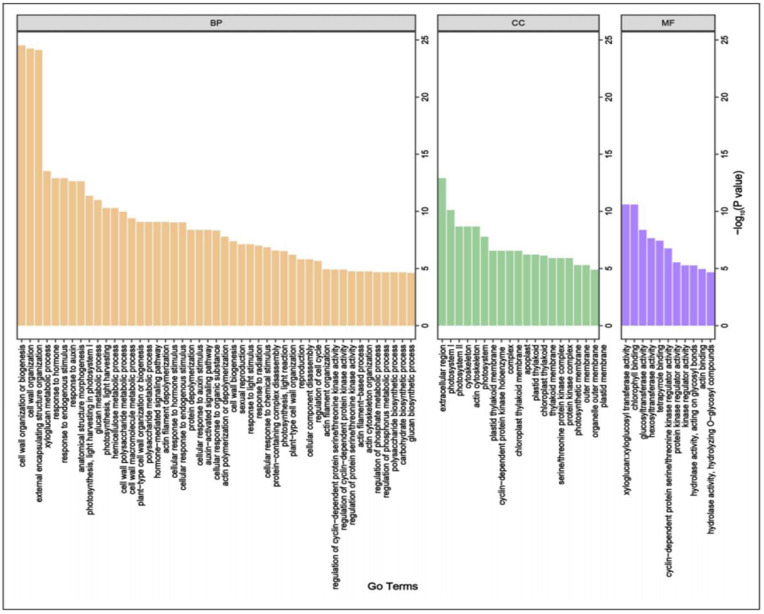
Gene ontology (GO) terms for the 188 candidate genes identified in the MQTL regions. BP: stands for biological processes, CC: stands for cellular components, MF: stands for molecular functions.

**Figure 5 plants-14-01943-f005:**
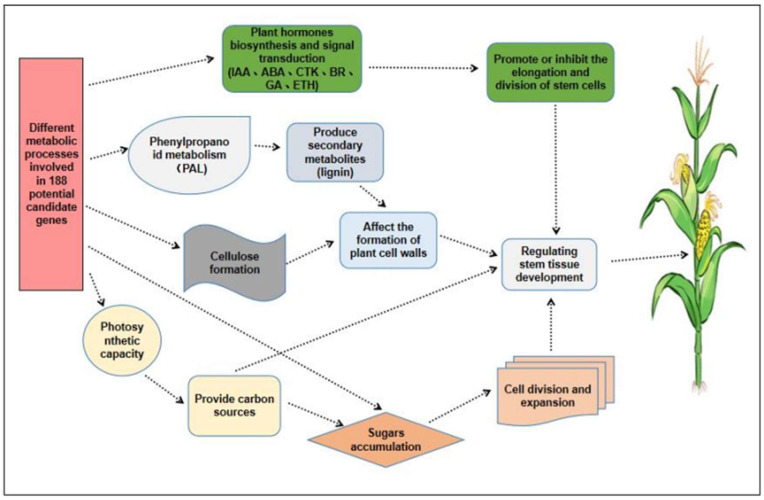
Molecular networks related to PH and EH development in maize.

**Table 1 plants-14-01943-t001:** QTL location information of maize PH and EH.

Population						Population Phenotypes				QTL Number		
Cross Group	Type	Size	Env.	Marker	Length (cM)	Range of PH (cm)	Range of EH (cm)	Mean PH (cm)	Mean EH (cm)	PH	EH	Reference
MO17 × SDM	F_2_	–	1	180/SSR	985.71	104.00–285.00	34.00–116.00	230.70	79.97	3	2	[23]
MU6 × SDM	F_2_	–	1	150/SSR	1085.81	21.00–290.00	34.00–116.00	224.10	74.85	3	3	
178 × 9782	RIL	271	1	259/SSR	1440.00	128.65–224.76	45.90–99.49	172.88	70.27	4	6	[24]
JB × Y53	F_2_	–	1	154/SSR	1735.00	96.00–211.00	18.00–75.00	173.53	50.31	3	10	[25]
	F_2:3_	211	1			130.40–210.38	27.00–95.78	168.41	62.62	6	11	
Chuan-287 × Chuan-144	F_2_	187	1	152/SSR	1268.10	172.25–306.00	54.50–144.56	241.02	90.53	3	3	[26]
JS037 × JS133	F_2_	192	1	136/SSR	1148.40	143.10–245.90	38.20–107.70	196.27	70.71	4	1	[27]
Chuan-287 × Chuan-144	F_3_	187	2	152/SSR	1268.10	142.25–256.23	44.50–124.56	172.20	65.88	3	3	[28]
						156.81–285.23	35.40–125.82	206.74	65.38	1	1	
	F_4_	187	1			151.42–274.81	39.56–121.33	202.99	68.57	3	3	
ms39ms39 × B73	F_2_	120	1	71/Indel,109/SSR	1173.40	134.80–273.00	41.10–101.40	–	–	4	2	[29]
T32 × HuangC	F_2:3_	184	3	193/SSR	1103.36	84.00–206.33	24.67–87.00	63.16	59.58	2	2	[30]
						103.23–201.72	33.67–84.28	161.70	57.93	3	3	
						112.38–207.41	28.58–90.73	160.98	57.10	2	3	
NX110 × NX531	DH	162	4	178/SSR	1721.19	121.00–224.00	32.00–99.67	170.82	64.39	1	2	[31]
						128.60–247.40	33.40–114.80	183.41	73.01	1	3	
						122.00–231.67	37.33–109.67	170.24	65.95	1	3	
						124.00–274.40	49.40–131.20	196.67	82.57	1	2	
YXD053 × Y6-1	RIL	202	2	200/SSR,12/AFLP	1648.60	109.83–163.67	–	136.84	–	3		[32]
						91.02–188.98	–	143.14	–	3		
Langhuang × TS141	F_2:3_	202	4	213/SSR	1542.50	135.00–261.73	49.32–138.52	204.61	94.27	3	3	[33]
						121.82–251.62	35.20–130.33	182.62	78.53	3	3	
						156.00–264.00	62.39–148.26	214.57	103.12	2	5	
						154.00–261.00	46.00–123.33	202.84	83.37	3	3	
Chang7-2 × TS141	F_2:3_	218	4	217/SSR	1648.80	100.28–51.21	50.00–165.30	171.75	94.24	4	5	[33]
						84.80–241.27	34.20–146.60	157.36	89.33	2	4	
						97.20–261.06	33.20–131.53	161.36	76.19	3	5	
						89.70–244.56	30.10–126.10	152.22	62.66	2	4	
B73 × Zheng58	RIL	165	1	189/SSR	2058.80	106.80–297.30	35.00–120.00	207.10	71.80	5	6	[34]
S112 × H132	F_2:3_	217	5	171/SSR	4734.51	135.00–300.00	43.00–179.00	227.67	100.26	1	1	[4]
						143.00–297.00	55.00–145.00	225.81	96.00	1	1	
						155.00–345.00	53.00–136.00	232.35	92.59	1	1	
						157.00–300.00	55.00–136.00	225.32	93.70	–	1	
						157.00–292.00	59.00–165.00	229.96	96.91	1	3	

Env.: environment; PH: plant height; EH: ear height; RIL: recombinant inbred line; DH: double haploid.

**Table 2 plants-14-01943-t002:** Results of meta-analysis of maize plant height and ear height.

Trait	MQTL	Chr.	Position (cM)	QTLs Number	Bin	Marker Interval	CI	Physical Interval (Mb)	Contig
PH, EH	MQTL1–1	1	30.90	6	1.01	umc2546–umc1292	5.40–59.20	2.09–5.41	ctg2–ctg3
PH, EH	MQTL1–2	1	130.30	2	1.02	bnlg1178–bnlg1429	125.00–143.50	14.07–16.56	ctg6–ctg7
PH, EH	MQTL1–3	1	232.50	5	1.03	umc1073–bnlg1203	208.60–259.30	32.87–43.71	ctg11
EH	MQTL1–4	1	402.70	2	1.04	bnlg2295–umc1243	398.20–405.00	80.17–83.54	ctg20
PH	MQTL1–5	1	488.10	3	1.05–1.06	umc1603–umc1972	475.90–503.30	165.42–178.04	ctg33–ctg38
PH, EH	MQTL1–6	1	779.40	4	1.08	bnlg1643–umc1991	768.50–800.70	232.80–245.37	ctg50
PH, EH	MQTL2–1	2	54.90	4	2.01–2.02	umc1542–umc1227	54.5–55.30	4.67–4.71	ctg69
PH, EH	MQTL2–2	2	89.10	5	2.02	umc1961–mmc0111	88.90–90.20	8.02–8.30	ctg70
PH, EH	MQTL2–3	2	404.00	3	2.07	bnlg1329–umc2129	383.5–403.40	184.75–188.81	ctg98
PH, EH	MQTL3–1	3	108.40	6	3.02–3.04	bnlg1647–bnlg1904	102.50–126.50	8.21–9.81	ctg112
PH, EH	MQTL3–2	3	193.90	10	3.04	umc1655–umc1504	189.90–227.40	26.31–58.49	ctg116–ctg119
PH, EH	MQTL3–3	3	350.10	8	3.04–3.05	umc1839–umc1087	343.50–364.60	155.29–161.90	ctg117–ctg131
PH, EH	MQTL3–4	3	474.70	3	3.06	umc1644–umc2269	472.30–478.00	183.94–184.73	ctg132–ctg138
PH, EH	MQTL3–5	3	567.20	7	3.07	umc1489–umc1286	566.80–569.60	202.17–202.87	ctg142–ctg143
PH, EH	MQTL3–6	3	803.50	3	3.09–3.10	umc1052–umc2048	789.00–817.20	226.86–230.27	ctg151–ctg151
EH	MQTL4–1	4	312.70	8	4.06	bnlg1741–bnlg1784	296.60–336.30	154.65–170.00	ctg182–ctg431
EH	MQTL4–2	4	535.10	4	4.09	umc1940–umc1650	524.30–544.60	221.18–230.89	ctg196–ctg200
PH, EH	MQTL5–1	5	586.90	9	5.06–5.07	umc2305–bnlg1346	459.20–534.40	193.36–208.04	ctg247–ctg251
EH	MQTL6–1	6	307.00	3	6.05	npi252–bnlg1702	304.00–312.70	143.66–145.85	ctg281–ctg285
PH, EH	MQTL7–1	7	187.20	6	7.02	umc1666–umc1932	181.10–202.50	47.95–78.03	ctg301–ctg308
PH, EH	MQTL7–2	7	401.80	5	7.03–7.04	umc2329–bnlg1666	382.10–428.20	151.28–158.98	ctg322–ctg323
EH	MQTL8–1	8	264.90	3	8.03–8.04	umc1457–umc1858	257.80–285.60	101.65–112.06	ctg345–ctg349
PH, EH	MQTL8–2	8	348.40	3	8.05	bnlg666–umc2210	340.20–366.30	133.37–152.00	ctg354–ctg358
PH	MQTL8–3	8	505.00	3	8.07–8.08	umc1384–bnlg1056	482.40–523.60	169.21–171.33	ctg363–ctg365
PH, EH	MQTL9–1	9	254.10	3	9.03–9.04	bnlg1626–bnlg1209	230.20–268.20	88.14–109.64	ctg376
PH, EH	MQTL10–1	10	68.70	7	10.01–10.02	umc1319–umc1432	47.40–91.40	4.61–5.77	ctg392
PH	MQTL10–2	10	144.90	2	10.03	umc1863–bnlg210	139.40–166.80	13.34–26.78	ctg394–ctg398
PH, EH	MQTL10–3	10	200.40	3	10.03–10.04	bnlg1079–umc1938	196.70–204.50	63.83–77.35	ctg400–ctg402
EH	MQTL10–4	10	232.10	2	10.03–10.04	umc2348–umc1077	229.40–236.90	93.28–102.71	ctg409–ctg411

Chr.: Chromosome; CI: Confidence interval.

## Data Availability

Data are contained within the article and Appendix A.

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
