# Peer review of "Meta-QTL Analysis and Genes Responsible for Plant and Ear Height in Maize (Zea mays L.)"

_plants, 2025, doi:10.3390/plants14131943_

Round 1
Reviewer 1 Report
Comments and Suggestions for Authors
Dear authors,
I carefully reviewed your manuscript and found and certain points need to be improved. Please consider the following comments:
Abstract:
1. The objective is not clearly and concisely defined. Clearly state the purpose, e.g., “This study aims to identify stable genomic regions and candidate genes associated with plant height (PH) and ear height (EH) in maize through meta-QTL analysis.”
2. Line 13: “A total of 187 original QTLs… 152 projected” – how and why 35 QTLs were not projected should be briefly addressed or acknowledged. Explain if these were omitted due to inconsistency, low resolution, or insufficient marker data.
Introduction:
1. The introduction provides background but does not clearly highlight the specific knowledge gap this study addresses.
2. The QTL examples are informative but lack synthesis. Multiple studies are listed, but no summary of common findings or challenges is provided.
3. It briefly mentions that MQTL analysis helps refine QTL regions, but doesn’t elaborate on the scientific significance of lower confidence intervals, or how this aids marker-assisted selection (MAS).
Results:
1. Many grammatical errors and typos impact readability.
Examples:
· “respnsible” → “responsible”
· “form these population” → “from these populations”
· “pontential” → “potential”
· “plasticy” → “plasticity”
· “theses candidate genes” → “these candidate genes”
· Perform careful proofreading and grammar correction throughout the results. It’s also advisable to use tools like Grammarly or ask a native speaker/editor for polishing.
2. Figures and tables are referenced but not described in sufficient depth.
3. LOD and PVE values are given for individual QTLs, but the criteria for significance (e.g., threshold LOD score) is not explained.
4. The number of MQTLs with candidate genes (27 out of 29) should include speculation why two lacked genes, e.g., due to gaps in genome annotation.
Discussion:
1. Emphasizes polygenic nature of PH and EH. Cites genes like VT2, SPI1, BR2, etc., with correct roles.
2. Good use of candidate gene functional annotation (hormones, cellulose, photosynthesis). Focuses on biologically meaningful candidates.
3. Statistical validation of MQTLs (confidence intervals, resolution gain vs. original QTLs) should be discussed briefly. Comparison with previously reported MQTL studies (if available) would add more value. Clearly differentiate “MQTL regions” from “candidate gene lists.”
Materials and Methods:
1. For QTLs: Clarify whether manual curation or automated scripts were used. Explicitly mention inclusion/exclusion criteria for selecting QTL studies.
2. Describe whether data extracted from multiple sources were cross-validated to avoid redundancy or inconsistencies.
3. The formula for calculating CI based on Darvasi and Soller (1997) should include a full citation and a brief rationale for using it.
4. Explain how QTLs from original studies were projected onto the consensus map — whether via interpolation, anchor markers, or software alignment.
5. Clarify if marker positions or trait values were standardized or normalized before projection.
6. Although AIC is mentioned, you should specify the range of models tested and how the optimal one was chosen.
7. Explain how candidate genes were filtered from the MQTL regions. What criteria determined “genes related to plant development”?
8. Indicate which gene annotation release/version (e.g., B73 RefGen_v4 or _v5) was used for mapping and gene retrieval.
Conclusion:
1. The conclusion should briefly summarize the significance of the 29 MQTLs and 188 genes. For example: Were these MQTLs located in known hotspots? How many QTLs per MQTL on average?
2. There is no mention of experimental validation.
3. Correct phrasing errors:
§ “photosynthesis capacity” → “photosynthetic capacity”
§ “corresponding candidate genes discovered in this study thus can be used…” → “The MQTLs and their corresponding candidate genes identified in this study can thus be used…”
Comments on the Quality of English LanguageDear authors,
Please proofread your manuscript for the language clarity. There are several typo and grammatical issues.
Author Response
Dear Editor and Reviewers
Thank you for your letter of – and for the referee’s comments concerning our manuscript, “Meta-QTL Analysis and Genes Responsible for Plant and Ear Height in Maize (Zea mays L.) (Manuscript ID: plants-3692346)”. We have carefully studied these comments and have made corresponding corrections to the manuscript, which we describe in detail below. We would like to re-submit the manuscript and that for possible publication on the Special Issue: “Genetic Diversity and Population Structure of Plants” of Plants. Thank you very much for your time and consideration.
Editor:
Your manuscript has now been reviewed by experts in the field and can be found with the review reports at: https://susy.mdpi.com/user/manuscripts/resubmit/2a7cd20059952e659056fbbb8c5b2bbb Please revise the manuscript found at the above link according to the reviewers' comments and upload the revised file within 10 days.
Thanks for the positive comments of you and all reviewers for our manuscript. As suggested, we have carefully revised and improved our manuscript using the “Track Changes” function of the manuscript at the above link. We then have re-submitted the manuscript within the allotted time.
Thank you for your consideration.
(I) Ensure all references are relevant to the content of the manuscript.
Thanks for the positive comments. As suggested, we have carefully checked all references. We then have re-submitted the manuscript.
Thank you for your consideration.
(II) Highlight any revisions to the manuscript, so editors and reviewers can see any changes made.
Thanks for the positive comments. As suggested, we have carefully revised and improved our manuscript using the “Track Changes” function of the manuscript. We then have re-submitted the manuscript.
Thank you for your consideration.
(III) Provide a cover letter to respond to the reviewers’ comments and explain, point by point, the details of the manuscript revisions.
Thanks for your positive comments for our manuscript. As suggested, we have carefully revised and improved our manuscript. In addition, we have prepared a detailed response letter to all reviewers for each point, and then have re-submitted the manuscript.
Thank you for your consideration.
(IV) If the reviewer(s) recommended references, critically analyze them to ensure that their inclusion would enhance your manuscript. If you believe these references are unnecessary, you should not include them.
Thanks for your positive comments for our manuscript. As suggested, we have carefully checked and revised the References. At the same time, we also have re-added twenty-one new references to enhance the quality of our manuscript. We then have re-submitted the manuscript.
Thank you for your consideration.
(V) If you found it impossible to address certain comments in the review reports, include an explanation in your appeal.
Thanks for your positive comments for our manuscript. As suggested, we have carefully revised and improved our manuscript. In addition, we have prepared a detailed response letter to all reviewers for each point, and then have re-submitted the manuscript.
Thank you for your consideration.
If your manuscript requires improvement to the language and/or figures, you may consider MDPI Author Services: https://www.mdpi.com/authors/english. Please note the status of this invitation “Publish Author Biography on the webpage of the paper” - https://susy.mdpi.com/user/manuscript/author_biography/b0aa56b7bef0a99dd4fe8c1bf66a47d6. If you wish to publish your biography, please complete it before your manuscript is accepted.
Thanks for the positive comments. As suggested, we have carefully checked and revised the English language of the manuscript. We then re-submitted the manuscript.
In addition, thanks for your invitation, we decided not to publish our biography.
Thank you for your consideration.
Please do not hesitate to contact us if you have any questions regarding the revision of your manuscript or if you need more time. We look forward to hearing from you soon.
Thanks for your positive comments for our manuscript. As suggested, we have carefully revised and improved the manuscript using the “Track Changes” function of our manuscript at the above link. We then have re-submitted the manuscript within the allotted time.
Thank you for your consideration.
Reviewer 1
Comments and Suggestions for Authors
I carefully reviewed your manuscript and found and certain points need to be improved. Please consider the following comments:
Thanks for your positive comments. As suggested, we modified the corresponding content. We then re-submitted the manuscript.
Thank you for your consideration.
Abstract:
- The objective is not clearly and concisely defined. Clearly state the purpose, e.g., “This study aims to identify stable genomic regions and candidate genes associated with plant height (PH) and ear height (EH) in maize through meta-QTL analysis.”
Thanks for your comments. We are very sorry for the inconvenience. As suggested, we modified the corresponding content were that “This study aims to identify stable genomic regions and candidate genes associated with PH and EH in maize through meta-QTL (MQTL) analysis.”in Lines 18-19 on page 1 of the manuscript. We then have re-submitted the manuscript.
Thank you for your consideration.
- Line 13: “A total of 187 original QTLs… 152 projected” – how and why 35 QTLs were not projected should be briefly addressed or acknowledged. Explain if these were omitted due to inconsistency, low resolution, or insufficient marker data.
Thanks for your positive comments. As suggested, we added the corresponding content were that “The remaining 35 QTLs could not be projected onto the consistency map, which may be attributed to a lack of common markers between the original and consistency map or to the QTL exhibiting low phenotypic variance explained (PVE), resulting in large confidence intervals (CIs).” in Lines 24-27 on page 1 of the manuscript. We then have re-submitted the manuscript.
Thank you for your consideration.
Introduction:
- 1. The introduction provides background but does not clearly highlight the specific knowledge gap this study addresses.
Thanks for your positive comments. As suggested, we modified the corresponding content were that “the aims of present study were to integrate multiple previously published QTLs related to PH and EH in maize, identify crucial genomic regions controlling these traits through MQTL analysis, obtain the corresponding MQTLs, and predict candidate genes.” in Lines111-1114 on page 3 of the manuscript. We then have re-submitted the manuscript.
Thank you for your consideration.
- 2. The QTL examples are informative but lack synthesis. Multiple studies are listed, but no summary of common findings or challenges is provided.
Thanks for your positive comments. As suggested, we added the corresponding content were that “However, due to differences in the size of mapping populations, genetic background, and selected markers, the same QTL is rarely found in different studies, which severely restrict the application potential of QTLs in maize molecular marker-assisted selection (MAS) breeding.” and “It can be seen from this that the use of MQTL analysis can help us determine the candidate genes related to maize PH and EH development.” in Lines 81-84 and 106-108 on page 2 and 3 of the manuscript. We then have re-submitted the manuscript.
Thank you for your consideration.
- 3. It briefly mentions that MQTL analysis helps refine QTL regions, but doesn’t elaborate on the scientific significance of lower confidence intervals, or how this aids marker-assisted selection (MAS).
Thanks for your positive comments. As suggested, we added the corresponding content were that “Among them, a lower CI can improve the positioning accuracy, reduce the number of candidate genes and enhance the reliability of the results.” in Lines 92-93 on page 2 of the manuscript. We then have re-submitted the manuscript.
Thank you for your consideration.
Results:
1.Many grammatical errors and typos impact readability. Examples: “respnsible” → “responsible”, “form theses population” → “from these populations”, “pontential” → “potential”、“plasticy” → “plasticity”, “theses candidate genes” → “these candidate genes”.
Perform careful proofreading and grammar correction throughout the results. It’s also advisable to use tools like Grammarly or ask a native speaker/editor for polishing.
Thanks for your positive comments. As suggested, we modified the errors listed above and conducted careful proofreading and grammar correction throughout the result We then have re-submitted the manuscript.
Thank you for your consideration.
- Figures and tables are referenced but not described in sufficient depth
Thanks for your positive comments. As suggested, we added the corresponding content were that in the results section. We then have re-submitted the manuscript.
Thank you for your consideration.
- LOD and PVE values are given for individual QTLs, but the criteria for significance (e.g., threshold LOD score) is not explained.
Thanks for your positive comments. As suggested, we added the corresponding content were that “Moreover, there only were six PH-QTLs and four EH-QTLs displayed high LOD threshold (LOD > 10), respectively (Figure 1B), and only 24.59% PH-QTLs and 32.62% EH-QTLs showed high PVE, respectively (Figure 1C). It is thus necessary to identify novel major QTLs with high values of LOD and PVE in future.” in Lines 142-1145 on page 3 of the manuscript. We then have re-submitted the manuscript.
Thank you for your consideration.
- The number of MQTLs with candidate genes (27 out of 29) should include speculation why two lacked genes, e.g., due to gaps in genome annotation.
Thanks for your positive comments. As suggested, we added the corresponding content were that “no candidate genes in MQTL2-1 and MQTL2-2,(Figure 3), this might be the result caused by a false positive original QTL mapping from previous experiments” in Lines 193-194 on page 8 of the manuscript. We then have re-submitted the manuscript.
Thank you for your consideration.
Discussion:
- Emphasizes polygenic nature of PH and EH. Cites genes like VT2, SPI1, BR2, etc., with correct roles.
Thanks for your positive comments. Here, the main purpose of citing genes such as VT2, SPI1 and BR2 is to illustrate that some genes related to the development of PH and EH in maize have been identified in current studies, and they are related to the biosynthesis and signal transduction of auxin, rather than emphasizing the polygenicity of PH and EH. We then have re-submitted the manuscript.
Thank you for your consideration.
- Good use of candidate gene functional annotation (hormones, cellulose, photosynthesis). Focuses on biologically meaningful candidates.
Thanks for your positive comments. In the identification of candidate genes, we made full use of the functional annotations of the candidate genes for identification. Eventually, based on the functional annotations of the candidate genes, 188 candidate genes related to the growth and development of maize plants were screened out, and these genes were involved in different metabolic processes. We then have re-submitted the manuscript.
Thank you for your consideration.
- Statistical validation of MQTLs (confidence intervals, resolution gain vs. original QTLs) should be discussed briefly. Comparison with previously reported MQTL studies (if available) would add more value. Clearly differentiate “MQTL regions” from “candidate gene lists.”
Thanks for your positive comments. As suggested, we modified the corresponding content were that “MQTL analysis is a method that can integrate the initial QTL data from different studies for a more comprehensive analysis. Meta-analysis can break through the limitations of various factors and solve the difficulties that it is difficult to locate all the target QTLs in a single QTL localization and the estimated PVE of QTL is often unreliable..”in Lines 255-259 on page 11 of the manuscript. We then have re-submitted the manuscript.
Thank you for your consideration.
Materials and Methods:
- For QTLs: Clarify whether manual curation or automated scripts were used. Explicitly mention inclusion/exclusion criteria for selecting QTL studies.
Thanks for your positive comments. As suggested, we added the corresponding content were that “The collected QTL were subjected to a quality check and QTL with PVE < 1%, or LOD < 1.5 were removed.” in Lines 326-327 on page 12 of the manuscript. We then have re-submitted the manuscript.
Thank you for your consideration.
- Describe whether data extracted from multiple sources were cross-validated to avoid redundancy or inconsistencies.
Thanks for your positive comments. The data we extracted from multiple sources did not find duplicate data due to the differences in the methods used, the size of the population and the parents. We only excluded QTLS with LOD<1.5 or PVE<1%, so no cross-validation of the data was conducted. We then have re-submitted the manuscript.
Thank you for your consideration.
- The formula for calculating CI based on Darvasi and Soller (1997) should include a full citation and a brief rationale for using it.
Thanks for your positive comments. As suggested, we added the corresponding content were that “The collected QTL were subjected to a quality check and QTL with PVE < 1%, or LOD < 1.5 were removed. Among them, the CI and PVE represented two fundamental parameters for QTL characterization. The meta-analysis of QTL was mainly achieved through the QTL LOD score, PVE, position, and CI. When the original QTL CI was missing, we can calculate the 95% CI based on the population type, size and PVE, with the equation provided by Darvasi and Soller (1997) [57].” in Lines 328-331 on page 12 of the manuscript. We then have re-submitted the manuscript.
Thank you for your consideration.
- Explain how QTLs from original studies were projected onto the consensus map — whether via interpolation, anchor markers, or software alignment.
Thanks for your positive comments. As suggested, we modified the corresponding content were that “In this study, we compared the original maps of QTLs with the IBM2 2008 Neighbors reference map. Next, we import all the information of these QTLs, especially the starting and ending positions of CI, into the program and map them to their own linkage maps. Then, based on their relative distances from the flanking framework markers, they are further projected onto the reference map while maintaining a constant distance between the frame marks. Then, the homogeneous functions provided by the BioMercator (V4.2) software were used to transfer each QTL to the reference map to create a consensus map of QTL related to maize PH and EH.” in Lines 343-351 on page 12 of the manuscript. We then have re-submitted the manuscript.
Thank you for your consideration.
- Clarify if marker positions or trait values were standardized or normalized before projection.
Thanks for your positive comments. As suggested, we modified the corresponding content were that “If the order of the side labels of the given QTL is reversed from that of the reference map, it has no effect on the projection of the QTL. If multiple QTLs are linked to the same marker, the CI is calculated using the formula proposed by Darvasi and Soller[16]. Otherwise, QTL is not predicted. ” in Lines 351-354 on page 11 of the manuscript. We then re-submitted the manuscript.
Thank you for your consideration.
- Although AIC is mentioned, you should specify the range of models tested and how the optimal one was chosen.
Thanks for your positive comments. As suggested, we modified the corresponding content that “Specifically, in accordance with the software specification and using the integrated CI data of QTL, we conducted a separate meta-analysis of PH-/ EH-QTL using the chromosome walk method. Furthermore, when conducting meta-analysis, first based on the optimal model values, i.e., AIC (akaike information content), AICc (AIC correction), AIC3 (AIC 3 candidate models), BIC (bayesian information criterion), and AWE (average weight of evidence) were used to determine the number of potential MQTL on each chromosome in different experiments. The QTL model with the lowest value among at least three of the five models was used to determine the number of MQTLS on each chromosome [59-60]. Finally, the location and CI of the existence of MQTL were estimated by using the maximum likelihood ratio of Gauss's theorem.” in Lines 366-375 on page 13 of the manuscript .We then re-submitted the manuscript.
Thank you for your consideration.
- Explain how candidate genes were filtered from the MQTL regions. What criteria determined “genes related to plant development”?
Thanks for your positive comments. As suggested, we added the corresponding content were that “Specifically, we used the "qTeller" tool provided on maizeGDB to select the expression dataset of B73 RefGen_v4 for plant development only, in order to identify the genes existing within the MQTL physical intervals [61].” in Lines 392-395 on page 13 of the manuscript. We then re-submitted the manuscript.
Thank you for your consideration.
- Indicate which gene annotation release/version (e.g., B73 RefGen_v4 or _v5) was used for mapping and gene retrieval.
Thanks for your positive comments. As suggested, we added the corresponding content in Lines 393 on page 13 of the manuscript .We then re-submitted the manuscript.
Thank you for your consideration.
Conclusion:
- The conclusion should briefly summarize the significance of the 29 MQTLs and 188 genes. For example: Were these MQTLs located in known hotspots? How many QTLs per MQTL on average?
Thanks for your positive comments. As suggested, we added the corresponding content were that “meanwhile each MQTL interval contained 2-10 original QTLs. Among them, the largest number of MQTL was detected on chromosomes 1 and 3, indicating that these two chromosomes may be important genomic regions controlling plant growth and development..” in Lines 401-404 on page 13 of the manuscript. We then re-submitted the manuscript.
Thank you for your consideration.
- There is no mention of experimental validation.
Thanks for your positive comments. As suggested, Later in the study, we will be according to the recognition of these candidate genes, from maizeEMSDB website (http://maizeems.qlnu.edu.cn/login/index.htm) to search for the genes related corn mutant materials, then the field planting and phenotypic observation, Finally, samples were taken for qRT-PCR analysis. We then re-submitted the manuscript.
Thank you for your consideration.
- Correct phrasing errors:
“photosynthesis capacity” → “photosynthetic capacity”
corresponding candidate genes discovered in this study thus can be used…” → “The MQTLs and their corresponding candidate genes identified in this study can thus be used...
Thanks for your positive comments. As suggested, We modified the corresponding content in Lines 411-412 on page 13 of the manuscript. We then re-submitted the manuscript.
Thank you for your consideration.
Best wishes!
Xiaoqiang Zhao Professor
State Key Laboratory of Aridland Crop Science, Gansu Agricultural University
- mail: zhaoxiaoq@gsau.edu.cn

Reviewer 2 Report
Comments and Suggestions for Authors
This study conducted a meta-QTL analysis using 187 QTLs from 13 studies to identify key genomic regions associated with plant height (PH) and ear height (EH) in maize. It identified 29 meta-QTLs and 188 candidate genes related to plant development, with GO analysis revealing involvement in 77 biological functions. The findings offer valuable insights for breeding maize varieties with optimal PH and EH for dense planting and improved yield.
Below are the comments.
Line 29-31; How does the study address the balance between high planting density and environmental stress tolerance in maize breeding?
Line 56-58; Were the QTLs identified in bin1.05, bin5.04-5.05, and bin6.04-6.05 consistent across different environments or trials?
Line 68-69; What criteria were used in the MQTL analysis to determine "lower confidence intervals" and "greater influence"?
Line 90-91; How did you ensure that the selection of the 13 original studies was unbiased and comprehensive? What were the inclusion and exclusion criteria for these studies?
Line 98-132; Could you clarify whether genotype-by-environment interactions were considered in the QTL analysis? How consistent were QTLs across environments? Were any of the QTLs identified as pleiotropic or co-localized for both PH and EH traits, and how were such overlaps analyzed? Some MQTLs still have large confidence intervals. What threshold did you consider acceptable for precise gene discovery, and how did you handle MQTLs with >30 cM intervals?
Line 152-153; What could be the reason for the absence of candidate genes in these two MQTL regions? Is it due to a lack of gene annotation in these regions, or were the intervals poorly defined?
Line 221-222;The authors speculate that PAL is involved in lignin formation. Is there transcriptomic or proteomic evidence to support the expression of Zm00001d010247 in stem tissues during critical stages of internode elongation?
Line 232-234; The identification of sugar-related genes in various MQTLs is interesting, but have the authors examined any phenotypic correlations between these loci and actual sugar content or transport efficiency in maize tissues?
Line 243-246; The role of ZmCRY1b in photomorphogenesis is cited. Was this gene identified directly from the MQTL region, and do the authors have evidence that its expression is spatially or temporally associated with stem development?
Line 254-255; MADS-box genes are said to regulate various developmental processes. Are these specific genes previously known to influence maize stem architecture, or is this an extrapolation from studies in other species?
Author Response
Dear Editor and Reviewers
Thank you for your letter of – and for the referee’s comments concerning our manuscript, “Meta-QTL Analysis and Genes Responsible for Plant and Ear Height in Maize (Zea mays L.) (Manuscript ID: plants-3692346)”. We have carefully studied these comments and have made corresponding corrections to the manuscript, which we describe in detail below. We would like to re-submit the manuscript and that for possible publication on the Special Issue: “Genetic Diversity and Population Structure of Plants” of Plants. Thank you very much for your time and consideration.
Editor:
Your manuscript has now been reviewed by experts in the field and can be found with the review reports at: https://susy.mdpi.com/user/manuscripts/resubmit/2a7cd20059952e659056fbbb8c5b2bbb Please revise the manuscript found at the above link according to the reviewers' comments and upload the revised file within 10 days.
Thanks for the positive comments of you and all reviewers for our manuscript. As suggested, we have carefully revised and improved our manuscript using the “Track Changes” function of the manuscript at the above link. We then have re-submitted the manuscript within the allotted time.
Thank you for your consideration.
(I) Ensure all references are relevant to the content of the manuscript.
Thanks for the positive comments. As suggested, we have carefully checked all references. We then have re-submitted the manuscript.
Thank you for your consideration.
(II) Highlight any revisions to the manuscript, so editors and reviewers can see any changes made.
Thanks for the positive comments. As suggested, we have carefully revised and improved our manuscript using the “Track Changes” function of the manuscript. We then have re-submitted the manuscript.
Thank you for your consideration.
(III) Provide a cover letter to respond to the reviewers’ comments and explain, point by point, the details of the manuscript revisions.
Thanks for your positive comments for our manuscript. As suggested, we have carefully revised and improved our manuscript. In addition, we have prepared a detailed response letter to all reviewers for each point, and then have re-submitted the manuscript.
Thank you for your consideration.
(IV) If the reviewer(s) recommended references, critically analyze them to ensure that their inclusion would enhance your manuscript. If you believe these references are unnecessary, you should not include them.
Thanks for your positive comments for our manuscript. As suggested, we have carefully checked and revised the References. At the same time, we also have re-added twenty-one new references to enhance the quality of our manuscript. We then have re-submitted the manuscript.
Thank you for your consideration.
(V) If you found it impossible to address certain comments in the review reports, include an explanation in your appeal.
Thanks for your positive comments for our manuscript. As suggested, we have carefully revised and improved our manuscript. In addition, we have prepared a detailed response letter to all reviewers for each point, and then have re-submitted the manuscript.
Thank you for your consideration.
If your manuscript requires improvement to the language and/or figures, you may consider MDPI Author Services: https://www.mdpi.com/authors/english. Please note the status of this invitation “Publish Author Biography on the webpage of the paper” - https://susy.mdpi.com/user/manuscript/author_biography/b0aa56b7bef0a99dd4fe8c1bf66a47d6. If you wish to publish your biography, please complete it before your manuscript is accepted.
Thanks for the positive comments. As suggested, we have carefully checked and revised the English language of the manuscript. We then re-submitted the manuscript.
In addition, thanks for your invitation, we decided not to publish our biography.
Thank you for your consideration.
Please do not hesitate to contact us if you have any questions regarding the revision of your manuscript or if you need more time. We look forward to hearing from you soon.
Thanks for your positive comments for our manuscript. As suggested, we have carefully revised and improved the manuscript using the “Track Changes” function of our manuscript at the above link. We then have re-submitted the manuscript within the allotted time.
Thank you for your consideration.
Reviewer 2
Comments and Suggestions for Authors
This study conducted a meta-QTL analysis using 187 QTLs from 13 studies to identify key genomic regions associated with plant height (PH) and ear height (EH) in maize. It identified 29 meta-QTLs and 188 candidate genes related to plant development, with GO analysis revealing involvement in 77 biological functions. The findings offer valuable insights for breeding maize varieties with optimal PH and EH for dense planting and improved yield.
Below are the comments.
Thanks for your positive comments. As suggested, we modified the corresponding content. We then re-submitted the manuscript.
Thank you for your consideration.
- Line 29-31; How does the study address the balance between high planting density and environmental stress tolerance in maize breeding?
Thanks for your positive comments. As suggested, In this study, we aim to identify candidate genes related to the development of corn plants through MQTL analysis. Then, based on the identified candidate genes, in the subsequent experiments, we will introduce the candidate genes into corn plants through gene editing technology to cultivate corn varieties with appropriate plant height and ear height, thereby solving the balance problem between high-density planting and environmental tolerance in maize breeding. We then re-submitted the manuscript.
Thank you for your consideration.
- Line 56-58; Were the QTLs identified in bin1.05, bin5.04-5.05, and bin6.04-6.05 consistent across different environments or trials?
Thanks for your positive comments. In different environments or experiments, the identified QTL in bin1.05, bin5.04-5.05 and bin6.04-6.05 is inconsistent. The marks intervals and control traits of the QTLs identified in these three bin intervals are not the same. We then re-submitted the manuscript.
Thank you for your consideration.
- Line 68-69; What criteria were used in the MQTL analysis to determine "lower confidence intervals" and "greater influence"?
Thanks for your positive comments. In MQTL analysis, we refer to the confidence interval with a genetic distance less than 5cM as a low confidence interval. For MQTLs that have a significant impact on the target trait, we can make a judgment based on the number of candidate genes identified within the interval. The more candidate genes identified within the interval, the greater the potential impact of the MQTL on the target trait. We then re-submitted the manuscript.
Thank you for your consideration.
- Line 90-91; How did you ensure that the selection of the 13 original studies was unbiased and comprehensive? What were the inclusion and exclusion criteria for these studies?
Thanks for your positive comments. To ensure the fairness and comprehensiveness of the selection of 13 original studies, we take the "maize", "PH", "EH", "QTL" as keywords, the NCBI (https://www.ncbi.nlm.nih.gov), CNKI (https://www.cnki.net), A systematic search was conducted in maizeGDB (http://www.maizegdb.org), and the corresponding QTL information of PH and EH was collected based on the original studies found. The original chromosome position, confidence interval (CI), phenotypic variation explanation (PVE), detailed level (LOD) value, parental name, population type, population size and marker type were included. Furthermore, we also conducted quality tests on the collected QTLS and removed QTLS with PVE < 1% or LOD < 1.5. Finally, because the 13 original research use map is different, we also use BioMercator v4.2.3 software (https://urgi.versailles.inra.fr/Tools/BioMercator-V4), and refer to the map IBM2 2008 neighbours, A comprehensive consensus graph has been constructed to ensure the reliability of the data. We then re-submitted the manuscript.
Thank you for your consideration.
- Line 98-132; Could you clarify whether genotype-by-environment interactions were considered in the QTL analysis? How consistent were QTLs across environments? Were any of the QTLs identified as pleiotropic or co-localized for both PH and EH traits, and how were such overlaps analyzed? Some MQTLs still have large confidence intervals. What threshold did you consider acceptable for precise gene discovery, and how did you handle MQTLs with >30 cM intervals?
Thanks for your positive comments. In the QTL analysis, the interaction between genotype and environment was taken into account. Among the 13 studies we collected, Zhao et al. [33] analyzed and found that QTLs with the same marker interval were detected under different moisture conditions. Thus, it can be seen that these QTLs are consistent. In the study of Fei et al. [4], it was found that the QTL controlling plant height and spike height was located in the umc2265-umc1839 interval. Meanwhile, Zhang et al. [34] also discovered the QTL that jointly controlled plant height and spike height in the bnlg1035-umc1644 range. Therefore, we believe that these QTLS have multiple effects from one cause. For precise gene discovery, we believe that the genetic distance of each MQTL interval should be less than 1.0cM (DOI: 10.1002/tpg2.20336). Furthermore, for MQTL with a confidence region >30cM, in the subsequent research, we need to further increase the density of tags for more precise positioning. We then re-submitted the manuscript.
Thank you for your consideration.
- Line 152-153; What could be the reason for the absence of candidate genes in these two MQTL regions? Is it due to a lack of gene annotation in these regions, or were the intervals poorly defined?
Thanks for your positive comments. As suggested, we added the corresponding content were that “(no candidate genes in MQTL2-1 and MQTL2-2,(Figure 3), this might be the result caused by a false positive original QTL mapping from previous experiments.)” in Lines 184-185 on page 8 of the manuscript. We then have re-submitted the manuscript.
Thank you for your consideration.
- Line 221-222;The authors speculate that PAL is involved in lignin formation. Is there transcriptomic or proteomic evidence to support the expression of Zm00001d010247 in stem tissues during critical stages of internode elongation?
Thanks for your positive comments. We are very sorry. Since Zm00001d010247 was initially identified by us in the MQTL interval, there is no verification of the transcriptomic or proteomic expression of this gene at the key stage of internode elongation in stem tissue here. But in Zhao et al. (DOI: 10.1016 / J.Y GENO. 2021.08.020) studies have found that PAL is a key enzyme in the process of lignin synthesis. Appropriate lignin can maintain the stability of the cell wall. However, if the content is too high, it will limit the elongation of the cells. Since the development of corn stems is closely related to cell elongation, it can be inferred that this gene may be related to the development of maize stems. We then re-submitted the manuscript.
Thank you for your consideration.
- Line 232-234; The identification of sugar-related genes in various MQTLs is interesting, but have the authors examined any phenotypic correlations between these loci and actual sugar content or transport efficiency in maize tissues?
Thanks for your positive commentsThank you for your consideration. We are very sorry about this. In this study, we have not verified any phenotypic correlations between these loci and the actual sugar content or transport efficiency in maize tissues. However, in the subsequent experiments, we will detect the expression differences of candidate genes in high and low sugar materials through qRT-PCR. We then re-submitted the manuscript.
Thank you for your consideration.
- Line 243-246; The role of ZmCRY1b in photomorphogenesis is cited. Was this gene identified directly from the MQTL region, and do the authors have evidence that its expression is spatially or temporally associated with stem development?
Thanks for your positive comments. The ZmCRY1b gene was not identified by us from the MQTL region. What we identified was the CRY1 gene. We cited the role of ZmCRY1b in photomorphogenesis to explain the effect of CRY on the development of maize plants. Studies have found that the maize overexpression line of ZmCRY1b inhibits the yellowing reaction of maize by enhancing the protein accumulation of bZIP transcription factors ZmHY5 and ZmHY5L. However, ZmHY5 and ZmHY5L can bind to the promoter region of GA oxidase ZmGA2ox10, leading to an upregulation of its transcriptional expression level, thereby reducing the decrease in active GA content in maize and further inhibiting plant height and ear position [53]. From this, we infer that the CRY1 gene is related to the development of corn plants. We then re-submitted the manuscript.
Thank you for your consideration.
- Line 254-255; MADS-box genes are said to regulate various developmental processes. Are these specific genes previously known to influence maize stem architecture, or is this an extrapolation from studies in other species?
Thanks for your positive comments. In Zhao et al. (DOI: 10.1080/21645698.2024.2328384) study found that ZmMADS42 gene expression in the corn stem tip meristem quantity highest, while these specific candidate genes in this study are our preliminary screening by MADS-box into the family genes. In view of the above research, we infer that these genes may be related to the development of maize stems. Therefore, in the subsequent research, we will verify the expression patterns of these specific genes in the target tissues through qRT-PCR. We then re-submitted the manuscript.
Thank you for your consideration.
Best wishes!
Xiaoqiang Zhao Professor
State Key Laboratory of Aridland Crop Science, Gansu Agricultural University
- mail: zhaoxiaoq@gsau.edu.cn

Reviewer 3 Report
Comments and Suggestions for Authors
The current paper devoted to meta-QTL analysis and identification of possible genes responsible for height of whole plants and ear in Zea Maize.
The topic is potentialy interesting , but text require corrections.
Below some details:
TITLE: please, improve:
“Genes Identification for Plant Height and Ear Height“ = „Genes responsible for plant and ear height“ as possible variant. But you can proposed better, of course.
Line 8: „significantly associated“?
Line 9: “dense planting characteristic and lodging resistance of maize” ¿? What is dense planting characteristics?
Lines 10- 11: please, edit sentence.
Line 27: „In recent years, high yield has always“ ??? In recent year or always? I think it is not only recent years..
Line 33: „hm-2“??
Lines 34- 37: 32 MQTLs located in different environmental conditions32 MQTLs located in different environmental conditionsthors can not mechanistically compared planting density: day length, temperature regime, soil type, nutrition – all factors should be consider. Under „optimal“ conditions denisty can be higher, while under poor nutrion, less sun light etc it can be lower to get highest possible productivity. +
Line 45: „PH and EH are important factors affecting the structure of maize plants [11].“ ???? PH and EH is not a factor, not a primary, but only are characterstic of plants.
Line 46: “appropriate PH and EH can enhance planting density” ????? How characteristics can enhace density??
Line 74: „32 MQTLs located in different environmental conditions“ ??? How MQTL can located in different envinronment? I see what did you mean, but current sentence require corrections, as well as many others to be directly understandable…
Line 82: „the genetic mechanisms of PH and EH“ ??? There are no „genetic mechanisms of PH and EH“ itself.
Line 99: „form“ = from???
Line 164: „metabolic progresses“ ??? = metabolic processes.
IAA – maize have 17 IAA synthesis gene located in different cell types and regulated different processes – root growth, mesophyll cell size, stomata, meristem size etc. Total IAA have a quite low biological relevance.
Line 166: „metabolism of phenylpropanoid, cellulose synthesis, sugars accumulation, cell division and growth (approximately for 32.62%) ?? it seems you combine here many opposite process as cell division, cell elongation, carbon fixation etc.. What is biological relevance of such combination? I know, many researchers do this mechanistic combination, but what is relevance in your opinion??
Line 179: „With rapid population growth, how to increase“ ?? = „With rapid population growth, the demand for increasing“ or something like this…
Line 181: „only 6.3 t/hm2 „ ?? of what? Total biomass? Grain? Etc? Moreover, you can not compare two different region with different soil type, different nutrions, different day length etc for productivity, plant density etc. Not gene, but gene expressions (regulated by conditions) is one of the key factor..
Line 188: „determine the tolerance of high-density maize cultivation“ ??? PH and EH is only a morphological characterstics and can not be considered as determination factor...
Line 193: “combined effects of additive and dominant effects” ?? Effects of effects??
Line 220: „imprtant role in development process of maize plants“ ??? important!
Line 222: lignin formation of plant stem [47] and thereby affects the growth of maize PH“ ????
Lignin itself is a secondary metabolites and can be only part of chain reactions invloved primary genes expression. There is not “growth of maize PH“…
Line 230: „Sugar not only serves as an energy source“??? Carbohydrates in plants are the main carbon source: key structural elements „responsible“ for PH and EH.
Line 241: „photosynthesis produces the carbohydrate, which provide energy for plant development [52]“ ???? Photosynthesis provide carbon source as a key building material.
Lines 313 – 314: „including six plant hormones biosynthesis and signal transduction, phenylpro- panoid metabolism, cellulose formation, sugars accumulation, cell division and expansion, photosynthesis capacity, and plant growth and development.“???
This is completely confusion point. Authors need to clearly distinquish between reason and consequence.
It will be nice to re-evaluate conclusion based on “butterfly effect” concept: relative low changes in expression of key genes lead to significant changes in “secondary genes” and strongly affected plant development.
Comments on the Quality of English LanguageMany sentences need corrections.
Author Response
Dear Editor and Reviewers
Thank you for your letter of – and for the referee’s comments concerning our manuscript, “Meta-QTL Analysis and Genes Responsible for Plant and Ear Height in Maize (Zea mays L.) (Manuscript ID: plants-3692346)”. We have carefully studied these comments and have made corresponding corrections to the manuscript, which we describe in detail below. We would like to re-submit the manuscript and that for possible publication on the Special Issue: “Genetic Diversity and Population Structure of Plants” of Plants. Thank you very much for your time and consideration.
Editor:
Your manuscript has now been reviewed by experts in the field and can be found with the review reports at: https://susy.mdpi.com/user/manuscripts/resubmit/2a7cd20059952e659056fbbb8c5b2bbb Please revise the manuscript found at the above link according to the reviewers' comments and upload the revised file within 10 days.
Thanks for the positive comments of you and all reviewers for our manuscript. As suggested, we have carefully revised and improved our manuscript using the “Track Changes” function of the manuscript at the above link. We then have re-submitted the manuscript within the allotted time.
Thank you for your consideration.
(I) Ensure all references are relevant to the content of the manuscript.
Thanks for the positive comments. As suggested, we have carefully checked all references. We then have re-submitted the manuscript.
Thank you for your consideration.
(II) Highlight any revisions to the manuscript, so editors and reviewers can see any changes made.
Thanks for the positive comments. As suggested, we have carefully revised and improved our manuscript using the “Track Changes” function of the manuscript. We then have re-submitted the manuscript.
Thank you for your consideration.
(III) Provide a cover letter to respond to the reviewers’ comments and explain, point by point, the details of the manuscript revisions.
Thanks for your positive comments for our manuscript. As suggested, we have carefully revised and improved our manuscript. In addition, we have prepared a detailed response letter to all reviewers for each point, and then have re-submitted the manuscript.
Thank you for your consideration.
(IV) If the reviewer(s) recommended references, critically analyze them to ensure that their inclusion would enhance your manuscript. If you believe these references are unnecessary, you should not include them.
Thanks for your positive comments for our manuscript. As suggested, we have carefully checked and revised the References. At the same time, we also have re-added twenty-one new references to enhance the quality of our manuscript. We then have re-submitted the manuscript.
Thank you for your consideration.
(V) If you found it impossible to address certain comments in the review reports, include an explanation in your appeal.
Thanks for your positive comments for our manuscript. As suggested, we have carefully revised and improved our manuscript. In addition, we have prepared a detailed response letter to all reviewers for each point, and then have re-submitted the manuscript.
Thank you for your consideration.
If your manuscript requires improvement to the language and/or figures, you may consider MDPI Author Services: https://www.mdpi.com/authors/english. Please note the status of this invitation “Publish Author Biography on the webpage of the paper” - https://susy.mdpi.com/user/manuscript/author_biography/b0aa56b7bef0a99dd4fe8c1bf66a47d6. If you wish to publish your biography, please complete it before your manuscript is accepted.
Thanks for the positive comments. As suggested, we have carefully checked and revised the English language of the manuscript. We then re-submitted the manuscript.
In addition, thanks for your invitation, we decided not to publish our biography.
Thank you for your consideration.
Please do not hesitate to contact us if you have any questions regarding the revision of your manuscript or if you need more time. We look forward to hearing from you soon.
Thanks for your positive comments for our manuscript. As suggested, we have carefully revised and improved the manuscript using the “Track Changes” function of our manuscript at the above link. We then have re-submitted the manuscript within the allotted time.
Thank you for your consideration.
Reviewer 3
Comments and Suggestions for Authors
The current paper devoted to meta-QTL analysis and identification of possible genes responsible for height of whole plants and ear in Zea Maize.
The topic is potentialy interesting , but text require corrections.
Thanks for your positive comments. As suggested, we modified the corresponding content. We then re-submitted the manuscript.
Thank you for your consideration.
- TITLE: please, improve:
“Genes Identification for Plant Height and Ear Height“ = „Genes responsible for plant and ear height“ as possible variant. But you can proposed better, of course.
Thanks for your positive comments. As suggested, we modified the corresponding content were that “Meta-QTL Analysis and Genes Responsible for Plant and Ear Height in Maize (Zea mays L.).” in Lines 2-3 on page 1 of the manuscript. We then have re-submitted the manuscript.
Thank you for your consideration.
- Line 8: “significantly associated”?
Thanks for your positive comments. As suggested, We modified the previous “significantly associated” to this “closely related to”. We then have re-submitted the manuscript.
Thank you for your consideration.
- Line 9: dense planting characteristic and lodging resistance of maize”? What is dense planting characteristics?
Thanks for your positive comments. As suggested, we added the corresponding content were that “Increasing the planting density will lead to changes in the structural characteristics of maize plants, such as reduced stem length and stem strength.” in Lines 12-13 on page 1 the of manuscript. We then have re-submitted the manuscript.
Thank you for your consideration.
- Lines 10- 11: please, edit sentence.
Thanks for your positive comments. As suggested, modified the corresponding content were that “Therefore, analyzing the genetic basis of PH and EH in maize can provide valuable information for cultivating ideal plant types with suitable PH and EH.” in Lines 14-16 on page 1 the of manuscript. We then have re-submitted the manuscript.
Thank you for your consideration.
- Line 27: „In recent years, high yield has always“ ??? In recent year or always? I think it is not only recent years..
Thanks for your positive comments. As suggested, we modified the corresponding content were that “increasing the yield of maize has become an important breeding goal.” We then have re-submitted the manuscript.” in Lines 39-40 on page 1 the of manuscript. We then have re-submitted the manuscript.
Thank you for your consideration.
- Line 33: “hm-2”??
Thanks for your positive comments. As suggested, we have modified “hm-2” to that “plants/hm2”. We then have re-submitted the manuscript.
Thank you for your consideration.
- Lines 34- 37: 32 MQTLs located in different environmental conditions32 MQTLs located in different environmental conditionsthors can not mechanistically compared planting density: day length, temperature regime, soil type, nutrition – all factors should be consider. Under „optimal“ conditions denisty can be higher, while under poor nutrion, less sun light etc it can be lower to get highest possible productivity. +
Thanks for your positive comments. As suggested, we added the corresponding content were that “Research has found that the yield of maize is affected by multiple factors such as the length of daylight, soil type, planting density, temperature and nutrient supply. Among them optimizing planting density serves as an effective strategy and a key approach to achieving consistently high yields in maize cultivation [3].”in Lines 43-45 on page 1 the of manuscript. We then have re-submitted the manuscript.
Thank you for your consideration.
- Line 45: PH and EH are important factors affecting the structure of maize plants [11].“ ???? PH and EH is not a factor, not a primary, but only are characterstic of plants.
Thanks for your positive comments. As suggested, we modified the corresponding content were that “Among them, PH and EH are the characteristic traits that affect the structure of maize plants [11].” in Lines 60-61 on page 2 of the manuscript. We then have re-submitted the manuscript.
Thank you for your consideration.
- Line 46: “appropriate PH and EH can enhance planting density” ????? How characteristics can enhace density??
Thanks for your positive comments. As suggested, we modified the corresponding content were that “appropriately reducing the PH and EH of maize can increase its planting density and yield” In Lines 62-63 on page 2 of the manuscript. We then have re-submitted the manuscript.
Thank you for your consideration.
- Line 74: „32 MQTLs located in different environmental conditions“ ??? How MQTL can located in different envinronment? I see what did you mean, but current sentence require corrections, as well as many others to be directly understandable…
Thanks for your positive comments. As suggested, we modified the corresponding content were that “In maize, Sheoran et al. [19] integrated original 542 QTLs from 33 published studies on tolerance to different abiotic stresses (drought, heat,salinity, cold, and waterlogging), who further performed MQTL analysis to obtain 32 MQTLs associated with multiple tolerance, and these MQTLs regions identified 1,907 poptential candidate genes.” in Lines 95-98 on page 2 of the manuscript. We then have re-submitted the manuscript.
Thank you for your consideration.
- Line 82: „the genetic mechanisms of PH and EH“ ??? There are no „genetic mechanisms of PH and EH“
Thanks for your positive comments. As suggested, we have modified “the genetic mechanisms of PH and EH” to that “ genetic basis of PH and EH.”. We then have re-submitted the manuscript.
Thank you for your consideration.
- Line 99: „form“ = from???
Thanks for your positive comments. As suggested, we have modified “form” to that“ from.” We then have re-submitted the manuscript.
Thank you for your consideration.
- Line 164: „metabolic progresses“ ??? = metabolic processes.
Thanks for your positive comments. As suggested, we have modified “metabolic progresses” to that “ metabolic processes.” We then have re-submitted the manuscript.
Thank you for your consideration.
- IAA – maize have 17 IAA synthesis gene located in different cell types and regulated different processes – root growth, mesophyll cell size, stomata, meristem size etc. Total IAA have a quite low biological relevance.
Thanks for your positive comments. In this study, we identified 19 genes related to the synthesis and signal transduction of IAA, among which 13 encoded auxin reactive protein (SAURs). Furthermore, through research, it was found that AtSAUR24 can promote the expansion of Arabidopsis thaliana cells, and the development of PH and EH in corn is closely related to the expansion of cells. Thus, it can be inferred that IAA may be related to the development of PH and EH. We then have re-submitted the manuscript.
Thank you for your consideration.
- Line 166: „metabolism of phenylpropanoid, cellulose synthesis, sugars accumulation, cell division and growth (approximately for 32.62%) ?? it seems you combine here many opposite process as cell division, cell elongation, carbon fixation etc.. What is biological relevance of such combination? I know, many researchers do this mechanistic combination, but what is relevance in your opinion??
Thanks for your positive comments. In my opinion, the relationship between them is as follows: The metabolism of phenylpropyl produces lignin (DOI: 10.1111/jipb.13054), and excessive accumulation of lignin will inhibit the elongation of cells. Cellulose is the main component of plant cell walls. It can increase the thickness and flexibility of stem cell walls [48], thereby enhancing the strength and anti-lodging ability of stems [49-50], and enabling the normal development of PH and EH. In addition, the accumulation of sugar can provide energy for cell activities and promote cell elongation. We then have re-submitted the manuscript.
Thank you for your consideration.
- Line 179: „With rapid population growth, how to increase“ ?? = „With rapid population growth, the demand for increasing“ or something like this…
Thanks for your positive comments. As suggested, we modified the corresponding content were that “With rapid population growth, people's demand for food is also constantly increasing. Therefore.” in Lines 224-225 on page 10 of the manuscript. We then have re-submitted the manuscript.
Thank you for your consideration.
- Line 181: „only 6.3 t/hm2 „ ?? of what? Total biomass? Grain? Etc? Moreover, you can not compare two different region with different soil type, different nutrions, different day length etc for productivity, plant density etc. Not gene, but gene expressions (regulated by conditions) is one of the key factor..
Thanks for your positive comments. As suggested, we modified the corresponding content were that “while its grain yield is only 6.3 t/hm2.”,Furthermore, we are conducting a comparison of grain production between the two regions here mainly to illustrate that the lower corn grain production in China compared to that in the United States is due to the low planting density. We then have re-submitted the manuscript.
Thank you for your consideration.
- Line 188: „determine the tolerance of high-density maize cultivation“ ??? PH and EH is only a morphological characterstics and can not be considered as determination factor...
Thanks for your positive comments. As suggested, we modified the corresponding content were that “PH and EH are the important agronomic traits closely related to the cultivation tolerance of high-density maize.” in Lines 233-235 on page 10 of the manuscript. We then have re-submitted the manuscript.
Thank you for your consideration.
- Line 193: “combined effects of additive and dominant effects” ?? Effects of effects??
Thanks for your positive comments. As suggested, we modified the corresponding content were that “the main genetic modes of PH and EH are additive and partial dominant effects.” in Lines 241-242 on page 10 of the manuscript. We then have re-submitted the manuscript.
Thank you for your consideration.
- Line 220: „imprtant role in development process of maize plants“ ??? important!
Thanks for your positive comments. As suggested, we have modified “imprtant” to that“ important.” We then have re-submitted the manuscript.
Thank you for your consideration.
- Line 222: lignin formation of plant stem [47] and thereby affects the growth of maize PH“ ????
Lignin itself is a secondary metabolites and can be only part of chain reactions invloved primary genes expression. There is not “growth of maize PH“…
Thanks for your positive comments. As suggested, we modified the corresponding content were that “which is related to the biosynthesis of lignin [47]. Appropriate lignin can maintain the stability of the cell wall, but if the content is too high, it may limit the elongation of cells, thereby affecting the growth of maize PH.” in Lines 276-279 on page 11 of the manuscript. We then have re-submitted the manuscript.
Thank you for your consideration.
- Line 230: „Sugar not only serves as an energy source“??? Carbohydrates in plants are the main carbon source: key structural elements „responsible“ for PH and EH.
Thanks for your positive comments. As suggested, we modified the corresponding content were that “Carbohydrates in plants are the main carbon source: key structural elements responsible for PH and EH.” in Lines 286-287 on page 11 of the manuscript. We then have re-submitted the manuscript.
Thank you for your consideration.
- Line 241: „photosynthesis produces the carbohydrate, which provide energy for plant development [52]“ ???? Photosynthesis provide carbon source as a key building material.
Thanks for your positive comments. As suggested, we modified the corresponding content were that “Photosynthesis provide carbon source as a key building material.” in Lines 300 on page 11 of the manuscript. We then have re-submitted the manuscript.
Thank you for your consideration.
- Lines 313 – 314: „including six plant hormones biosynthesis and signal transduction, phenylpro- panoid metabolism, cellulose formation, sugars accumulation, cell division and expansion, photosynthesis capacity, and plant growth and development.“???
This is completely confusion point. Authors need to clearly distinquish between reason and consequence.
It will be nice to re-evaluate conclusion based on “butterfly effect” concept: relative low changes in expression of key genes lead to significant changes in “secondary genes” and strongly affected plant development.
Thanks for your positive comments. As suggested, we modified the corresponding content were that “including six plant hormones biosynthesis and signal transduction, phenylpropanoid metabolism, cellulose formation, sugars accumulation, cell division and expansion, and photosynthetic capacity. Furthermore, based on the above results and previous studies, we constructed a possible molecular network related to the development of PH and EH in maize (Figure 5).”in Lines 406-411 on page 13 of the manuscript. We then have re-submitted the manuscript.
Thank you for your consideration.
Figure 5. Molecular networks related to PH and EH development in maize.
Best wishes!
Xiaoqiang Zhao Professor
State Key Laboratory of Aridland Crop Science, Gansu Agricultural University
- mail: zhaoxiaoq@gsau.edu.cn

Round 2
Reviewer 1 Report
Comments and Suggestions for Authors
The authors have addressed all my comments and concerns; however, I have a few minor corrections:
- Figure 5. I suggest explaining this figure in the results and discussions, and shifting this figure next to its explanation.
- Lines 330 - 334: Please explain equations 1 and 2, including the abbreviations used in them.
- Figure 4. Please include all sub-figures in the caption.
Author Response
Dear Editor and Reviewers
Thank you for your letter of – and for the referee’s comments concerning our manuscript, “Meta-QTL Analysis and Genes Responsible for Plant and Ear Height in Maize (Zea mays L.) (Manuscript ID: plants-3692346)”. We have carefully studied these comments and have made corresponding corrections to the manuscript, which we describe in detail below. We would like to re-submit the manuscript and that for possible publication on the Special Issue: “Genetic Diversity and Population Structure of Plants” of Plants. Thank you very much for your time and consideration.
Editor:
Your manuscript has now been reviewed by experts in the field and can be found with the review reports at: https://susy.mdpi.com/user/manuscripts/resubmit/2a7cd20059952e659056fbbb8c5b2bbb Please revise the manuscript found at the above link according to the reviewers' comments and upload the revised file within 10 days.
Thanks for the positive comments of you and all reviewers for our manuscript. As suggested, we have carefully revised and improved our manuscript using the “Track Changes” function of the manuscript at the above link. We then have re-submitted the manuscript within the allotted time.
Thank you for your consideration.
(I) Ensure all references are relevant to the content of the manuscript.
Thanks for the positive comments. As suggested, we have carefully checked all references. We then have re-submitted the manuscript.
Thank you for your consideration.
(II) Highlight any revisions to the manuscript, so editors and reviewers can see any changes made.
Thanks for the positive comments. As suggested, we have carefully revised and improved our manuscript using the “Track Changes” function of the manuscript. We then have re-submitted the manuscript.
Thank you for your consideration.
(III) Provide a cover letter to respond to the reviewers’ comments and explain, point by point, the details of the manuscript revisions.
Thanks for your positive comments for our manuscript. As suggested, we have carefully revised and improved our manuscript. In addition, we have prepared a detailed response letter to all reviewers for each point, and then have re-submitted the manuscript.
Thank you for your consideration.
(IV) If the reviewer(s) recommended references, critically analyze them to ensure that their inclusion would enhance your manuscript. If you believe these references are unnecessary, you should not include them.
Thanks for your positive comments for our manuscript. As suggested, we have carefully checked and revised the References. At the same time, we also have re-added twenty-one new references to enhance the quality of our manuscript. We then have re-submitted the manuscript.
Thank you for your consideration.
(V) If you found it impossible to address certain comments in the review reports, include an explanation in your appeal.
Thanks for your positive comments for our manuscript. As suggested, we have carefully revised and improved our manuscript. In addition, we have prepared a detailed response letter to all reviewers for each point, and then have re-submitted the manuscript.
Thank you for your consideration.
If your manuscript requires improvement to the language and/or figures, you may consider MDPI Author Services: https://www.mdpi.com/authors/english. Please note the status of this invitation “Publish Author Biography on the webpage of the paper” - https://susy.mdpi.com/user/manuscript/author_biography/2a7cd20059952e659056fbbb8c5b2bbb. If you wish to publish your biography, please complete it before your manuscript is accepted.
Thanks for the positive comments. As suggested, we have carefully checked and revised the English language of the manuscript. We then re-submitted the manuscript.
In addition, thanks for your invitation, we decided not to publish our biography.
Thank you for your consideration.
Please do not hesitate to contact us if you have any questions regarding the revision of your manuscript or if you need more time. We look forward to hearing from you soon.
Thanks for your positive comments for our manuscript. As suggested, we have carefully revised and improved the manuscript using the “Track Changes” function of our manuscript at the above link. We then have re-submitted the manuscript within the allotted time.
Thank you for your consideration.
Reviewer 1
Comments and Suggestions for Authors
The authors have addressed all my comments and concerns; however, I have a few minor corrections:
Thanks for your positive comments. As suggested, we modified the corresponding content. We then re-submitted the manuscript.
Thank you for your consideration.
- Figure 5. I suggest explaining this figure in the results and discussions, and shifting this figure next to its explanation.
Thanks for your comments. As suggested, we modified the corresponding content were that “In addition, these metabolic processes interact with each other, thereby affecting the development process of corn plants (Figure 5).” and “In summary, we have tried to construct a molecular network that may regulate the PH and EH development in maize (Figure 5). Briefly, when maize plants are growing rapidly, multiple metabolic pathways controlled by related genes will be triggered. For example, when triggering metabolic pathways related to the synthesis and signal transduction of plant hormones, it will promote or inhibit the elongation of stem cells. When triggering pathways related to phenylpropanoid metabolism and cellulose formation, they will affect the formation of the cell wall. Ultimately, under the combined influence of these pathways, the stem development is affected, resulting in existing significance differences in PH and EH from various maize materials.” in Lines 196-197 and 301-309 on page 8 and 12 of the manuscript. We then have re-submitted the manuscript.
Thank you for your consideration.
- Lines 330 - 334: Please explain equations 1 and 2, including the abbreviations used in them.
Thanks for your comments. As suggested, we added the corresponding content were that “where CI was the confidence interval of a QTL, N was the size of the mapping population, and PVE was the phenotypic variance explained by QTL. Equation (1) was applicable to backcross and F2 populations. Equation (2) was applied to RIL populations.” in Lines 329-331 on page 13 of the manuscript. We then have re-submitted the manuscript.
Thank you for your consideration.
- Figure 4. Please include all sub-figures in the caption.
Thanks for your comments. As suggested, we added the corresponding content were that “BP: stands for biological processes, CC: stands for cellular components, MF: stands for molecular functions.” in Lines 206-207 on page 10 of the manuscript. We then have re-submitted the manuscript.
Thank you for your consideration.
Reviewer 2
Comments and Suggestions for Authors
The manuscript has been sufficiently improved.
Thanks for your positive comments.
Reviewer 3
Comments and Suggestions for Authors
Thank you for response. The text still reuire minri revision.
Thanks for your positive comments. As suggested, we modified the corresponding content. We then re-submitted the manuscript.
Thank you for your consideration.
For example,
- Line 61: "appropriately reducing the PH and EH of maize can increase its planting density"?? PH and EH can not have effect on density!!
Thanks for your comments. As suggested, we modified the corresponding content were that “On the one hand, appropriately reducing the PH and EH of maize can increase its planting density and yield. Meanwhile, they can also improve the lodging resistance of maize and facilitate mechanical harvesting [12]. Moreover, excessive PH and EH are important limiting factors for maize dense corn planting.” in Lines 55-58 on page 2 of the manuscript. We then have re-submitted the manuscript.
Thank you for your consideration.
- Line 197: "188 candidate genes" - how did authors consider differential "gene value" and localization? genes responsible for hormonal production have definitely "more valöue-effect" as some "secondary metabolism" genes. I hope you will consider it in the future work.
Thanks for your comments. Screening for MQTL range from 188 candidate genes, in subsequent studies we will from qTeller website (https://qteller.maizegdb.org) to download a variety of corn organization transcriptome data, Then, the expression characteristics of the candidate genes identified in the main stages and different organizations were analyzed using the public database qTeller. Furthermore, in our future work, we will also focus on considering the genes responsible for hormone production.We then have re-submitted the manuscript.
Thank you for your consideration.
- Moreover, it looks like that authors overlooked the role of auxin (IAA/IPA) in PH and EH regulation. In the future it will be great to clarify which auxin biosynthesis gene (form 17 in maize) responsible for PH and which for EH. But authors can mentioned this point in discussion in the current text. You can look here: "https://www.tandfonline.com/doi/full/10.1080/15592324.2021.1891756".
Thanks for your comments. As suggested, we added the corresponding content were that “However, as an important signaling substance regulating the growth and development of maize, the synthesis and transduction mechanisms of IAA have not been fully clarified [47]. In this study, we identified 18 candidate genes related to IAA synthesis and signal transduction associated with the development of PH and EH in maize. This discovery can provide certain references for the subsequent clarification of the synthesis and transduction mechanisms of IAA.” in Lines 256-261 on page 11 of the manuscript. We then have re-submitted the manuscript.
Thank you for your consideration.
Best wishes!
Xiaoqiang Zhao Professor
State Key Laboratory of Aridland Crop Science, Gansu Agricultural University
- mail: zhaoxiaoq@gsau.edu.cn

Reviewer 2 Report
Comments and Suggestions for Authors
The manuscript has been sufficiently improved.
Author Response

(The authors gave the same response as above.)

Reviewer 3 Report
Comments and Suggestions for Authors
Thank you for response. The text still reuire minri revision.
For example,
Line 61: "appropriately reducing the PH and EH of maize can increase its planting density"?? PH and EH can not have effect on density!!
Line 197: "188 candidate genes" - how did authors consider differential "gene value" and localization? genes responsible for hormonal production have definitely "more valöue-effect" as some "secondary metabolism" genes. I hope you will consider it in the future work.
Moreover, it looks like that authors overlooked the role of auxin (IAA/IPA) in PH and EH regulation. In the future it will be great to clarify which auxin biosynthesis gene (form 17 in maize) responsible for PH and which for EH. But authors can mentioned this point in discussion in the current text. You can look here: "https://www.tandfonline.com/doi/full/10.1080/15592324.2021.1891756".
My best regards!
Author Response

(The authors gave the same response as above.)
